# Influence of 2000-2050 climate change on particulate matter in the United States: Results from a new statistical model

Lu Shen[1], Loretta J. Mickley[1], Lee T. Murray[2]

[1]School of Engineering and Applied Sciences, Harvard University, Cambridge, MA 02138, USA

[2]Department of Earth and Environmental Sciences, University of Rochester, Rochester, NY 14627, USA

*Correspondence to*: Lu Shen (lshen@fas.harvard.edu)

**Abstract.** We use a statistical model to investigate the effect of 2000-2050 climate change on fine particulate matter ($PM_{2.5}$) air quality across the contiguous United States. By applying observed relationships of $PM_{2.5}$ and meteorology to the IPCC Coupled Model Intercomparision Project Phase 5 (CMIP5) archives, we bypass some of the uncertainties inherent in

chemistry-climate models. Our approach uses both the relationships between $PM_{2.5}$ and local meteorology as well as the synoptic circulation patterns, defined as the Singular Value Decomposition (SVD) pattern of the spatial correlations between $PM_{2.5}$ and meteorological variables in the surrounding region. Using an ensemble of 19 GCMs under the RCP4.5 scenario, we project an increase of 0.4-1.4 $\mu g \ m^{-3}$ in annual mean $PM_{2.5}$ in the eastern US and a decrease of 0.3-1.2 $\mu g \ m^{-3}$ in the Intermountain West by the 2050s, assuming present-day anthropogenic sources of $PM_{2.5}$. Mean summertime $PM_{2.5}$ increases

as much as 2-3 $\mu g \ m^{-3}$ in the eastern United States due to faster oxidation rates and greater mass of organic aerosols from biogenic emissions. Mean wintertime $PM_{2.5}$ decreases by 0.3-3 $\mu g \ m^{-3}$ over most regions in United States, likely due to the volatilization of ammonium nitrate. Our approach provides an efficient method to calculate the climate penalty or benefit on air quality across a range of models and scenarios. We find that current atmospheric chemistry models may underestimate or even fail to capture the strongly positive sensitivity of monthly mean $PM_{2.5}$ to temperature in the eastern United States in

summer, and may underestimate future changes in $PM_{2.5}$ in a warmer climate. In GEOS-Chem, the underestimate in monthly mean $PM_{2.5}$-temperature relationship in the East in summer is likely caused by overly strong negative sensitivity of monthly mean low cloud fraction to temperature in the assimilated meteorology ($\sim$-0.04 $K^{-1}$), compared to the weak sensitivity implied by satellite observations ($\pm$0.01 $K^{-1}$). The strong negative dependence of low cloud cover on temperature, in turn, causes the modeled rates of sulfate aqueous oxidation to diminish too rapidly as temperatures rise, leading to the

underestimate of sulfate-temperature slopes, especially in the South. Our work underscores the importance of evaluating the sensitivity of $PM_{2.5}$ to its key controlling meteorological variables in climate-chemistry models on multiple timescales before they are applied to project future air quality.

# 1 Introduction

Fine particulate matter with an aerodynamic diameter less than 2.5 μm ($PM_{2.5}$) is an important surface air pollutant of public concern, particularly in industrialized regions. Exposure to $PM_{2.5}$ can result in respiratory and cardiovascular disease, as well as premature mortality (e.g., Laden et al., 2006; Pellucchi et al., 2009; Brook et al., 2010). In the United States, recent reductions in anthropogenic emissions have decreased $PM_{2.5}$ concentrations by 20% from 2001 to 2010 (EPA 2011; Hu et al., 2014), and this trend is very likely to continue in the future due to increasingly stringent emission control (Val Martin et al., 2015). However, a changing climate modifies local meteorological variables, synoptic circulation, and natural emissions, and thus brings new challenges to projections of future $PM_{2.5}$. $PM_{2.5}$ is comprised of a variety of individual components, including sulfate, nitrate, ammonium, organic carbon (OC) and elemental carbon (EC). The response of different $PM_{2.5}$ components to meteorology is complex (Tai et al., 2010), and model projections of $PM_{2.5}$ under the 21[st] century climate change have so far shown little consistency (e.g., Racherla and Adams, 2006; Pye et al., 2009; Val Martin et al., 2015; Day et al., 2015). In this study, we develop a new statistical model to quantify the effect of 2000 to 2050 climate change on $PM_{2.5}$ air quality across the contiguous United States.

The response of $PM_{2.5}$ to local meteorological variables differs by component, region, and time of year. Analyzing observations from across the United States, Tai et al. (2010) found that sulfate, OC, and elemental carbon increases with temperature everywhere due to faster oxidation rates, as well as the association of warmer temperatures with stagnation, reduced ventilation, and greater biogenic and fire emissions. Tai et al. (2010) also determined that the correlation of nitrate with temperature is negative in the Southeast but positive in California and the Great Plains due to the competing effects of temperature on emissions and condensation. These authors further found that higher relative humidity (RH) increases both sulfate, by enhancing in-cloud $SO_2$ oxidation, as well as nitrate due to the RH dependence of ammonium nitrate formation. Conversely, higher RH decreases OC and EC due to the association of moist air with reduced wildfires and greater influx of clean marine air (Tai et al., 2010). The relationship of $PM_{2.5}$ with clouds and precipitation is complex: as cloud cover increases, aqueous-phase oxidation of $SO_2$ increases, but greater precipitation may also scavenge all $PM_{2.5}$ components (Koch et al., 2003; Tai et al., 2010). These varied and sometimes competing effects of meteorology on the different components of $PM_{2.5}$ make it challenging to predict $PM_{2.5}$ variability.

Beside local meteorology, synoptic circulation patterns also play an important role in affecting $PM_{2.5}$ air quality. For example, Thishan Dharshana et al. (2010) found that synoptic weather systems contribute 30% of the $PM_{2.5}$ daily variability in the Midwestern United States. Tai et al. (2012a) found that 20-40% of the observed $PM_{2.5}$ daily variability can be explained by cold frontal passages in the eastern United States and maritime inflow in the West. But characterizing the effects of cold front passages and other synoptic patterns on surface $PM_{2.5}$ is challenging. Indices reflective of such patterns – e.g., the polar jet (Barnes and Fiore, 2013), cyclone frequency (Mickley et al., 2004; Leibensperger et al., 2008), and the extent of the

Bermuda High (Li et al., 2011; Shen et al., 2015) – may reflect only a fraction of the total synoptic activity in some regions, and the relationships between these patterns and $PM_{2.5}$ are not completely understood.

Chemical transport models (CTMs) and chemistry-climate models (CCMs) show no consistent sign of the future $PM_{2.5}$ changes under a changing climate (e.g., Liao et al., 2006; Racherla and Adams, 2006; Tagaris et al., 2007; Heald et al., 2008; Avise et al., 2009; Pye et al., 2009). Reviewing earlier studies, Jacob and Winner (2009) and Fiore et al. (2015) concluded that the most of the projected effects of 21$^{st}$ century climate changes on $PM_{2.5}$ concentrations are in the range of ±0.1-1 µg m$^{-3}$, with changes up to ±2 µg m$^{-3}$ in certain seasons or regions. More recently, Val Martin et al. (2015) found that 2000-2050 climate change may decrease the annual mean $PM_{2.5}$ concentrations by 0-1 µg m$^{-3}$ in the eastern United States under the Representative Concentration pathway (RCP) 4.5 scenario of climate change. Day et al. (2015) determined that summer mean $PM_{2.5}$ increases by 21% in the Southeast but decreases 9% in the Northeast from 2000 to 2050 under the more greenhouse-gas intensive A2 scenario. In contrast, Gonzalez-Abraham et al. (2015) identified a 10-30% increase of summer mean $PM_{2.5}$ across the eastern United States by the 2050s. A key reason for these inconsistencies is the large variation in the projections of future meteorology from climate models, regardless of scenario. Due to their high computation expense, CTMs typically rely on the meteorological fields from a single climate model. But the dependence of $PM_{2.5}$ on meteorological variables such as temperature is also uncertain, especially over longer timescales (e.g., interannual or decadal). To our knowledge, the ability of models to reproduce the dependence of $PM_{2.5}$ on major meteorological variables over such long timescales has not yet been evaluated.

An alternative approach to projecting the effect of climate change on $PM_{2.5}$ air quality involves the use of statistical models, in which the observed relationships of $PM_{2.5}$ and meteorology are applied to future climate projections from an ensemble of models. Use of an ensemble provides a mean or median response and uncertainty range and increases confidence in the sign and magnitude of the response of a particular variable to climate change. For example, Tai et al. (2012b) first analyzed 1999-2010 observations using principal component analysis of eight different meteorological variables, and found the interannual variability of $PM_{2.5}$ is strongly correlated with the average cyclone period $T$, defined as the inverse of the median frequency of the dominant meteorological mode, in the continuous United States. They then projected 2000 to 2050 changes in $PM_{2.5}$ by applying the local $PM_{2.5}$-to-period sensitivity (i.e., $\Delta (PM_{2.5})/\Delta T$) to the future changes in the average cyclone period $T$ derived from an ensemble of climate model simulations following the A1B scenario. Results showed only a weak increase of ~0.1 µgm$^{-3}$ in annual mean $PM_{2.5}$ in the eastern United States, and a likely weak decrease in the Pacific Northwest. However, Tai et al. (2012b) may have underestimated the change in future $PM_{2.5}$ because only the influence of synoptic patterns was considered and not the impact from local meteorology. More recently, Lecoeur et al. (2014) developed a statistical algorithm to estimate future $PM_{2.5}$ concentrations over Europe based on a weather-type representation. They resampled future daily $PM_{2.5}$ concentrations from a pool of chemistry model simulations, based on the similarity determined by regression-estimated

PM$_{2.5}$ and large-scale circulations. They found seasonal mean PM$_{2.5}$ changes between -1.6 and +1.1 μgm$^{-3}$ under RCP4.5 scenario by 2050s.

In this study, we revisit the conclusions of Tai et al. (2012b). We develop a new method to characterize the synoptic circulations using the Singular Value Decomposition (SVD) of the spatial correlations between PM$_{2.5}$ and meteorological variables in the surrounding region. The method takes into account the influence of both local meteorology and the synoptic circulation patterns to investigate the effect of 2000-2050 climate change PM$_{2.5}$ air quality across the contiguous United States. We also evaluate different CTMs and CCMs in terms of the simulated dependence of seasonal mean PM$_{2.5}$ on temperature over one decade. In Section 2, we introduce the data and models we use. In Section 3, the method used to characterize the synoptic circulation patterns is described. We discuss the projected 2000 to 2050 changes in PM$_{2.5}$ in Section 4. Section 5 evaluates the capability of different dynamic models in simulating the dependence of PM$_{2.5}$ on key meteorological variables.

## 2 Data sources and Models

### 2.1 PM$_{2.5}$ and meteorological data

Surface daily mean PM$_{2.5}$ concentrations and speciation data from 1999 to 2013 are taken from the U.S. Environmental Protection Agency Air Quality System (EPA-AQS, http://www.epa.gov/ttn/airs/airsaqs/). We interpolate the site measurements onto a 2.5°×2.5° latitude-by-longitude grid, using inverse distance weighting as in Tai et al. (2010). The meteorological data used in this study for 1999-2013 consist of temperature, relative humidity, and east-west and north-south wind speed from the National Centers for Environmental Prediction (NCEP) Reanalysis 1, mapped onto the 2.5°×2.5° grid resolution (Kalnay et al., 1996). For precipitation, we rely on the NOAA Climate Prediction Center (CPC) Unified Gauge-Based Analysis of Daily Precipitation product for 1999-2013 (Xie et al., 2007; Chen et al., 2008). These variables have been used previously to predict PM$_{2.5}$ (e.g., Tai et al., 2010, 2012a, 2012b; Lecœur et al., 2014), and their variability is closely linked to that of synoptic patterns (e.g., Shen et al., 2015; Thishan Dharshana et al., 2010). These particular variables have also been validated in CMIP5 models (e.g., Sheffield et al., 2012).

Satellite-observed cloud fractions for 2004-2012 are from the Clouds and the Earth's Radiant Energy System (CERES) ISCCP-D2like products (CERES Science Team, Hampton, VA, USA: NASA Atmospheric Science Data Center, accessed Oct, 2016, at http://doi.org/10.5067/Aqua/CERES/ISCCP-D2LIKE-MERG00_L3.003A). This merged product combines 3-hourly, daytime cloud properties from Terra and Aqua on the Moderate Resolution Imaging Spectroradiometer (MODIS) and from the geostationary satellite (GEO), mapped over the 1°×1° grid resolution (Minnis et al., 1995; 2011). The cloud optical depths are archived in three wavelength bins (0-3.6, 3.5-23, and 23-380 μm) in both liquid and ice phases. In this

study, we focus on clouds in the lower troposphere below 680 hPa, which have the greatest implications for surface $PM_{2.5}$ air quality.

To project the 2000-2050 effect of climate change on $PM_{2.5}$ air quality, we use five meteorological variables –surface temperature, relative humidity, precipitation, and east-west and north-south wind speed – from an ensemble of 19 climate models participating in the Coupled Model Intercomparison Project Phase 5 (CMIP5) and following the RCP 4.5 scenario (Taylor et al., 2012). RCP4.5 is an intermediate scenario, in which the radiative forcing reaches 4.5 W m$^{-2}$ by 2100, approximately 650 ppm $CO_2$ concentration, and stabilizes after that (Taylor et al., 2012). The CMIP5 data are archived at a horizontal resolution of ~200 km, and the details of these models can be found in Table S1.

To remove the effects of long-term trend, we subtract the 5-year moving average from monthly mean values in both $PM_{2.5}$ and meteorological data as in Tai et al. (2012b). The choice of five years is arbitrary, but we find this choice produces good correlations between surface $PM_{2.5}$ and meteorological variables over the relatively short 15-year $PM_{2.5}$ time history of observations, thus allowing us to bypass the impact of non-linear emission changes. Throughout this study, we use $p < 0.05$ as the threshold for statistical significance.

## 2.2 Atmospheric chemistry models

We preform a 9-year simulation of $PM_{2.5}$ in the GEOS-Chem CTM (v9-02, http://geos-chem.org) with coupled gas-phase and aerosol chemistry. The model has a horizontal resolution of 2°×2.5° with 47 pressure levels extending from surface to 0.01 hPa (~38 in the troposphere), driven by GEOS-5 assimilated meteorological data for 2004 to 2012 from the NASA Global Modeling and Assimilation System (GMAO). The aerosol thermodynamical partitioning of nitrate and ammonium between gas and aerosol phases is calculated by the ISORROPIA II model (Fountoukis and Nenes, 2007). The scheme to produce sulfate via aqueous-phase oxidation of $SO_2$ uses liquid water content and cloud fraction from the assimilated meteorology (Fisher et al., 2011). Formation of secondary organic aerosol (SOA) follows Pye et al. (2010), with many subsequent updates to the isoprene oxidation mechanism (Paulot et al., 2009a, b; Rollins et al., 2009). Biogenic emissions are from the inventory of *Guenther et al.* (2012). We follow Hudman et al. (2012) for emissions of nitrogen oxides ($NO_x$) from soil, and Murray et al. (2012) for lightning $NO_x$. U.S. anthropogenic emissions of $PM_{2.5}$ precursors are from the EPA 2005 National Emissions Inventory (NEI05). We use monthly biomass burning emissions from Global Fire Emission Database (GFED, van der Werf et al., 2010).

GEOS-5 assimilates a large array of observations but calculates clouds properties using a prognostic algorithm without assimilation. The algorithm considers both liquid and ice phases of cloud condensate with two types of cloud types, anvil and large-scale clouds (Reinecker et al., 2008). The basic moist processes include a convective scheme using the Relaxed Arakawa-Schubert parameterization (Moorthi and Suarez, 1992), a large-scale cloud condensate scheme (Smith, 1990,

Rotstayn, 1997), and cloud destruction schemes as described in (Reinecker et al., 2008). Column cloud fraction in the lower troposphere is calculated using a random overlap approximation (Stephens et al., 2004). In Section 5, we validate the GEOS-5 cloud fraction in the lower troposphere against CERES satellite observations.

Finally, we use modeled, 1995-2010 $PM_{2.5}$ surface concentrations and temperature data from the Atmospheric Chemistry and Climate Model Intercomparison Project (ACCMIP). For this historical simulation, the ACCMIP models follow the same time-varying anthropogenic and biomass burning emissions (Lamarque et al., 2010). Only four ACCMIP models provide archived total $PM_{2.5}$ concentrations: NCAR-CAM3.5, GFDL-AM3, MIROC-CHEM and GISS-E2-R (Table S2). Here we use an updated simulation with the GISS-ModelE2 model in its atmosphere-only mode, forced using the ACCMIP emissions
(Lamarque et al., 2010), observed daily sea-surface temperatures and sea-ice from Reynolds et al. (2007), and with winds nudged to the Modern-Era Retrospective Analysis for Research and Applications (MERRA) meteorological reanalysis (Rienecker et al., 2011). The rate constants for oxidation of $SO_2$ and DMS by OH have been updated to those recommended by Burkholder et al. (2015), consistent with GFDL-AM3 and GEOS-Chem. All four ACCMIP models are CCMs. The horizontal resolution of these models is ~200 km; more details are described in Lamarque et al. (2013).

**3 Construction of synoptic circulation factors**

$PM_{2.5}$ variability is not only related to local meteorology, but also synoptic circulation. Previous studies have identified many synoptic patterns that are important for surface air quality in different regions under certain seasons, such as cyclone frequency (Mickley et al., 2004; Leibensberger et al., 2008), the position of the polar jet wind in the Northeast (Barnes and Fiore, 2013; Shen et al., 2015), and the extent of the Bermuda High west edge in summer in the Southeast (Li et al., 2011;
Shen et al., 2015). However, identification and interpretation of the dominant synoptic patterns for each region and each month would be time consuming and subject to some uncertainty. Instead, as a first step, we attempt to find a more general way to characterize the major synoptic patterns that modulate the $PM_{2.5}$ variability.

Synoptic circulation plays a vital role in controlling $PM_{2.5}$ air quality. The correlations of surface $PM_{2.5}$ with meteorological
variables in the surrounding regions may in fact be stronger than those in the local regions. For example, Figure 1a shows the correlations between May-June-July (MJJ) monthly mean $PM_{2.5}$ concentrations in one $2.5° \times 2.5°$ grid box in Georgia in the southeast United States with MJJ surface air temperatures in grid boxes across a much larger domain ($32.5° \times 17.5°$) over the 1999-2013 time period. Positive correlations extend across the whole Southeast, suggesting that $PM_{2.5}$ air quality in Georgia is affected by regional climate; the strongest correlations are located in Mississippi, ~500 km west of Georgia. The
relationship of $PM_{2.5}$ in the Georgia gridbox with relative humidity also shows a regional signature, with negative correlations spanning the Southeast to the Gulf of Mexico (Figure 1b). Precipitation can scavenge particles, and we identify negative correlations of the Georgia $PM_{2.5}$ with regional precipitation (Figure 1c). The relationships of Georgia $PM_{2.5}$ with

east-west wind speed are relatively weak, with negative correlations in the Midwest and Gulf of Mexico (Figure 1d). However, the relationships of $PM_{2.5}$ in the Georgia grid box with the north-south wind speed show a strong bimodal structure, with significant negative correlations stretching over the eastern Atlantic and positive correlations in the south central United States (Figure 1e), suggesting anti-cyclonic circulation. In contrast the correlation of this variable with $PM_{2.5}$ within Georgia is close to zero, which means the local north-south wind speed does not provide predicative capability for $PM_{2.5}$ here. Taken together these results imply that $PM_{2.5}$ variability is partly controlled by regional-scale synoptic patterns, and consideration of only local meteorology will not suffice in predicting $PM_{2.5}$.

We construct the synoptic circulation factors driving $PM_{2.5}$ across the eastern United States through the use of SVDs of the spatial correlations between $PM_{2.5}$ in each grid box and meteorological variables in the surrounding region. This SVD method effectively compresses the information from several meteorological variables in a multi-dimensional matrix into a set of scalars that represent the oscillation of the $PM_{2.5}$-related synoptic patterns. For each grid box, the process proceeds as below. First, we calculate the correlations of monthly mean $PM_{2.5}$ in the grid box with five meteorological variables (temperature, relative humidity, precipitation, and north-south and west-east wind speed) within a ~1,000-km radius of the grid box on a 2.5°×2.5° horizontal grid. This step yields a 13×9×5 (longitude × latitude × variable) matrix which we call $A$. Second, we align the dimension of longitude-latitude into one column, and resize matrix $A$ into a 117×5 two-dimensional matrix $F$. The SVDs of $F$ can be written as

$$F = ULV^T$$

where $L$ is a diagonal matrix with non-negative numbers on the diagonal. Each column of $V$ represents the variable weights and each column of $U$ represents the spatial weights of the corresponding SVD mode. For example, Figure 2a-b shows the spatial and variable weights of the first SVD (SVD1) mode for $PM_{2.5}$ in the same gridbox in Georgia as in Figure 1, where SVD1 explains 32% of the total variance. The spatial weights show a bimodal structure with negative anomalies over the eastern Atlantic and positive anomalies over the Great Plains and Midwest (Figure 2a), in a pattern similar to that in Figure 1e. The corresponding variable weights in Figure 2b reveal the importance of the north-south wind speed in this mode, suggesting that SVD1 is characterized by dynamic, synoptic-scale meteorology. In the second SVD (SVD2) mode, the spatial weights (Figure 2c) show positive anomalies in the southeast United States, and this corresponds to the positive temperature anomalies in Figure 1a as well as negative relative humidity and precipitation anomalies in Figure 1b-c. The meteorological composition of the variable weights shows that temperature, relative humidity, and precipitation dominate (Figure 2d), suggesting that SVD2 reflects a regional-scale thermal effect. The magnitudes of SVD1 and SVD2 oscillate over time, contributing to $PM_{2.5}$ variability in the Georgia gridbox. We repeat this exercise for each grid box across the United States.

The magnitude of each $PM_{2.5}$-related mode in a new meteorological field can be calculated as follows. For each grid box, we first construct a matrix $M$, consisting of the monthly mean values of each meteorological variable across the surrounding

region. We scale the time series of each variable in each grid box to achieve zero mean and unit standard deviation across the time frame. The magnitude of each SVD mode for every month $t$ is then calculated using the inverse process of SVD, which can be written as

$$S_k = U_k^T M_t V_k$$

where $U_k$ refers to the $k^{th}$ column in the spatial weights matrix $U$, $V_k$ to the $k^{th}$ column in the variable weights matrix $V$, and $S_k$ is a scalar depicting the magnitude of the $k^{th}$ SVD mode of the new meteorological field for that month. This inverse SVD transforms a large matrix into a few scalars, and these scalars reflect the variability of synoptic patterns that are closely related to $PM_{2.5}$ air quality.

We first construct a multiple linear regression model to correlate observed monthly mean 1999-2013 $PM_{2.5}$ concentrations and five local meteorological variables (surface temperature, relative humidity, precipitation, and east-west wind and north-south wind) and the two most important synoptic factors in each gridbox, diagnosed using SVD. The model is of the form

$$Y = \sum_{k=1}^{5} \alpha_k X_k + \sum_{n=1}^{2} \beta_n S_n + b$$

where $Y$ is three continuous monthly mean $PM_{2.5}$ concentrations for 1999-2013 with a total number of 45 values in the time series. For example, for July $PM_{2.5}$, we train the model using June, July and August values for each year over the 15 years. $X$ is a scalar consisting of the five local meteorological variables, $S$ represents the two synoptic circulation factors constructed using SVD, $\alpha$ and $\beta$ are the corresponding coefficients, and $b$ is the intercept. We test this model in two steps. In the first step, we use only the local meteorological variables – i.e., we set all $\beta s$ to zeros. In the second step, we use both local meteorology and synoptic patterns. In order to avoid over-fitting, we use leave-one-out cross-validation to determine the best variable combinations for each gridbox. Each time we reserve one observation in the timeseries as the test set and use the remaining ones as the training set, and we repeat this process until all observations have been predicted. Throughout this study, we predict monthly $PM_{2.5}$ concentrations using this regression model, but projected changes of $PM_{2.5}$ in the future climate will be displayed as seasonal and annual means.

Figure 3a shows the cross-validated skills expressed in the coefficients of determination ($R^2$) between observed and predicted 1999-2013 monthly mean $PM_{2.5}$ concentrations using only local meteorology. We find $R^2$ averages 34% across the United States, with the largest $R^2$ located in the Midwest, Northeast and Northwest. This spatial pattern of $R^2$ is consistent with the pattern in Tai et al. (2010), who regressed daily $PM_{2.5}$ concentrations onto only local meteorological variables. By including synoptic circulation factors into the model, the average $R^2$ of the regression model increases over most regions, with an average $R^2$ across the United States of 43% and $R^2$ values greater than 50% over a broad region that includes the upper Midwest, Ohio, parts of the Northeast, and areas as far south as Tennessee (Figure 3b). This result demonstrates that inclusion of synoptic circulation factors can significantly improve the regression model. We also find that the cross-validated

values of $R^2$, calculated from both local meteorology and patterns of synoptic circulation and averaged across the United States, are 35% in spring, 44% in summer, 42% in autumn and 43% in winter (Figure S1). To check the multi-colinearity among predictors in this model, we calculate the variance inflation factors (VIFs) for all variables in each gridbox and each month. Results in Figure S2 show that about 98.9% of the VIFs are less than 5, well below the threshold of 10 that defines significant multi-colinearity (Kutner et al., 2004).

## 4 Impact of 2000-2050 climate changes on $PM_{2.5}$ from statistical inference

To estimate the impacts of climate change on future $PM_{2.5}$ concentrations from 2000-2019 to 2050-2069, we apply the regression model including both local and synoptic meteorology to the CMIP5 meteorological projections. We calculate mean surface $PM_{2.5}$ in both timeframes and then the resulting change. We assume that anthropogenic emissions of $PM_{2.5}$ sources remain at mean 1999-2013 levels during the 2050-2059 timeframe. An ensemble of 19 CMIP5 models in the RCP4.5 scenario is used here, and we calculate the $PM_{2.5}$ change for each model separately. Computing the average $PM_{2.5}$ change across the ensemble improves confidence in our predictions of the climate impact on $PM_{2.5}$.

Future climate change by 2050s leads to significant warming across North America, but has minimal effects on precipitation and circulation patterns across the continent. Figure S3 shows the seasonal changes in temperature, relative humidity, precipitation and surface wind field for June-July-August (JJA) across the United States, averaged across the CMIP5 ensemble. Mean temperature increases by 2-2.5 K over much of the North in this timeframe, and 1.5-2 K over the Southeast. Relative humidity decreases by up to 0.03 over most regions across the United States, but the models show no consistent sign in the future change in precipitation in the summer. The flux of maritime air into the south United States increases due to increased land-ocean thermal contrast. In winter (Figure S4), mean temperature increases by 3 K in the North, while relative humidity decreases across the intermountain west and the Northeast, similar to the pattern in summer. Precipitation shows a slight increase of 0.1 mm d$^{-1}$ in the North, and the surface circulation pattern shows little change over the United States (Figure S4).

Figure 4a-d shows the response of the seasonal mean $PM_{2.5}$ concentrations to 2050s climate change across the United States, shown as the average of all projections from the CMIP5 models. $PM_{2.5}$ increases by ~2-3 µg/m$^3$ in summer in the eastern United States (Figure 4b), likely due to faster oxidation rates and more abundant organic aerosol (OA) in the warmer climate of the 2050s (e.g., Tai et al, 2010; Kelly et al, 2012; Gonzalez-Abraham et al., 2015). This can be also inferred from the positive sensitivity of sulfate and OA with temperatures from observations, which will be discussed in more details in Section 5. We also find an increase of ~0.8-1.5 µg m$^{-3}$ in the summer over the Intermountain West, partly driven by enhanced biomass burning in a warmer climate (e.g., Yue et al., 2013, 2015). In winter, future $PM_{2.5}$ decreases by 0.3-3 µg/m$^3$ across much of the United States (Figure 4d), likely driven by greater volatilization of ammonium nitrate at warmer

temperatures (Dawson et al., 2007, 2009). In spring and autumn, PM$_{2.5}$ increases in the eastern United States by ~0.5 μg/m$^3$. Annual mean PM$_{2.5}$ increases as much as 1.4 μg/m$^3$ in the eastern United States but decreases by up to 1 μg/m$^3$ in the inter-mountain West (Figure 4e).

To evaluate the uncertainty of projected PM$_{2.5}$ concentrations, we analyze the range of these projections among the 19 CMIP5 models as well as the interannual timeseries of regional projections from 2000-2069. Even though many models have multiple simulations, when we calculate the effects of climate change on PM$_{2.5}$ concentrations, we only use the simulated meteorology from the first ensemble run for each model. In general, these models agree on the sign of the change of PM$_{2.5}$ across the East by 2050s, but the magnitude of the change varies among models (Figure S5). To more rigorously characterize

this uncertainty, we calculate the 90$^{th}$ and 10$^{th}$ percentile changes in PM$_{2.5}$ concentrations as calculated from the 19 CMIP5 models (Figure S6-S7). In the summertime, the 90$^{th}$ percentile changes of PM$_{2.5}$ can be greater than 3 μg/m$^3$ across most of the eastern United States (Figure S6b), but the 10$^{th}$ percentile changes are only 0.5-1.5 μg/m$^3$ (Figure S7b). These discrepancies underscore the importance of using an ensemble of climate models to project future PM$_{2.5}$ concentrations. Such an approach allows us to identify robust results across models, quantify uncertainty, and diagnose model outliers. We

also examine the 2000-2069 timeseries of projected PM$_{2.5}$ concentrations as annual, summertime, and wintertime means, averaged over eight different U.S. (Figure S8-11). The spread in PM$_{2.5}$ trends is one measure of the uncertainty in our projections, arising in part from differences in model sensitivity to changing greenhouse gases and in part from internal variability of the climate system (e.g., Deser et al., 2013). Averaging results across the CMIP5 ensemble reveals a robust response of PM$_{2.5}$ to increasing greenhouse gases, at least in some regions, giving us confidence in our approach.

    We also compare our results to those from recent studies using chemistry-climate models. Among the seven recent studies reviewed in Fiore et al (2015), only two of them projected a significant increase of PM$_{2.5}$ concentrations in summer over the eastern United States. Kelly et al. (2012) estimated an increase of 0.5-1.0 μg m$^{-3}$ in summertime PM$_{2.5}$ over much of the East from 2000 to 2050, mainly resulting from rapid increases in SOA from biogenic emissions. Gonzalez-Abraham et al. (2015)

found that the effect of 2000-2050 climate change alone without changes in biogenic emissions can increase PM$_{2.5}$ concentrations by up to 1.0 μg m$^{-3}$ in the eastern United States, a combined effect of increasing sulfate and ammonium as well as decreasing nitrate. Consideration of the changes in biogenic emissions drives up this increase to 0.5-3 μg m$^{-3}$.

    To diagnose which meteorological variable plays the greatest role in these PM$_{2.5}$ changes, we perform a series of tests with

the regression model. For each test, we keep one variable in the 2050-2069 calculation the same as for the 2000-2019 timeframe and calculate the resulting changes in PM$_{2.5}$. We find that the changes of PM$_{2.5}$ almost vanish if we hold surface temperatures for 2050-2069 at their 2000-2019 values (Figure S12), suggesting that temperature drives most of the PM$_{2.5}$ changes in the future climate.

Our study shows much larger regional effects of 2000-2050 climate change on annual mean $PM_{2.5}$ compared to Tai et al. (2012b). An increase of only ~0.1 μg/m$^3$ was predicted by Tai et al. (2012b) in the eastern United States, an order of magnitude smaller than what we find. We trace the reason for this discrepancy to the choice of predictors in the two studies. Tai et al. (2012b) identified the dominant meteorological modes driving daily $PM_{2.5}$ variability in 4°×5° gridcells across the United States and calculated the local sensitivity of $PM_{2.5}$ to synoptic period $T$ for that mode. Using the simulated changes in $T$ from a set of climate models, they then projected future $PM_{2.5}$ in each gridcell. Tai et al. (2012b) further found a strong correlation ($r = -0.63$) between $T$ and the maximum eddy growth rate, a quantity that reflects the meridional temperature gradient. This finding implies that trends in $T$ represent only the changes in the meridional temperature gradient, but do not take into account the effects of homogeneous warming across the mid and high latitudes. Partly to remedy this bias, we have included both local meteorology and synoptic circulations patterns into our regression model, leading to a much higher response of $PM_{2.5}$ to climate change.

One weakness of this study is that when estimating the sensitivity of $PM_{2.5}$ to meteorological variables, we do not consider the impact of changing anthropogenic emissions on this sensitivity. Figure S13 compares the slopes of monthly mean $PM_{2.5}$ and its components with temperature for two time periods: 1999-2006 summers with high anthropogenic emissions and 1997-2013 summers with low anthropogenic emissions. Using the monthly data, we find that the changes of sensitivity of $PM_{2.5}$ to temperature vary across different locations and species. As the anthropogenic emissions decrease, the slopes of $PM_{2.5}$ and temperature decrease over the Great Plains and Midwest, but increase slightly in the south Atlantic States. Sulfate exhibits decreased sensitivity across the eastern United States, and OA shows no significant pattern of change. Reasons for such inconsistencies may be related to the shorter time periods and therefore less robust sensitivity. In this study, we have thus chosen not to investigate the influence of changing emissions on the sensitivity of $PM_{2.5}$ to climate change using this statistical model.

## 5 Evaluation of $PM_{2.5}$ sensitivity to surface temperature in chemistry models

A key question is why previous model studies show no consistent sign in the in the change of future $PM_{2.5}$ relative to the present (Jacob and Winner, 2009). Such discrepancies no doubt arise in part because of differences in model projections of future climate or in model speciation of $PM_{2.5}$. In this section we investigate whether differences in model representation of the sensitivity of $PM_{2.5}$ to meteorological variability may also contribute to uncertainty in projections of future $PM_{2.5}$. As we point out above, few or no models have undergone evaluation of their capability in simulating this sensitivity over relatively long time scales (e.g., the interannual variability over a decade). Our tests with the regression model show that temperature is the most important driver of changing $PM_{2.5}$ in a changing climate, making it the primary candidate for evaluation in these models. We focus on summer (JJA) because our predictions point to an increase of $PM_{2.5}$ by 2050s of 2-3 μg m$^{-3}$ in the eastern United States by the 2050s at that time of year, values much greater than previous predictions.

This section consists of two parts. First, we test the capability of four ACCMIP models and GEOS-Chem in capturing the observed relationship between JJA monthly mean $PM_{2.5}$ and temperature. We find no model simulates this relationship well. Second, using GEOS-Chem as a testbed, we investigate the reasons of this failure in this particular model.

Figure 5 shows the distributions of the slopes of monthly $PM_{2.5}$ and temperature over the United States in observations and in different chemistry models for summer months in the present-day. All $PM_{2.5}$ and temperature values have been detrended, as described above, so that the slopes reflect only the $PM_{2.5}$ response to the interannual variability in temperature. For both the observations and the model results, the sensitivities of $PM_{2.5}$ to temperature shown here encapsulate the response of $PM_{2.5}$ to

all variables associated with temperature, including cloud cover, relative humidity, and boundary layer height. The observations display positive slopes over the whole United States, with slopes in the East greater than 1 µg m$^{-3}$ K$^{-1}$ (Figure 5a). The positive slopes driven by faster oxidation rates and increased biogenic emissions, as well as the stagnation frequently concurrent with higher temperatures. The models, however, either underestimate the positive slopes or even yield negative slopes in some regions, with no consistent spatial patterns in these discrepancies. For example, CAM3.5 shows

significant positive slopes in Texas, the Midwest, and Northeast (Figure 5b). GFDL-AM3 displays a bimodal structure, with positive slopes in the Northeast but negative slopes in the South (Figure 5c). The GISS-ModelE2 shows slight positive slopes over parts of the East (Figure 5d). The slopes in MIROC-CHEM are nearly flat, indicating little sensitivity of the monthly mean $PM_{2.5}$ concentrations to temperature variability (Figure 5e). GEOS-Chem shows positive slopes over much of the eastern United States, but the magnitudes are much less than those observed (Figure 5f). In a more recent study,

Westervelt et al. (2016) used a multivariate linear model to check the dependence of $PM_{2.5}$ on meteorology in the GFDL Coupled Model (CM3), and identified a positive $PM_{2.5}$-temperature sensitivity in the East in CM3 when all monthly data across the year were considered. For summer, however, Westervelt et al. (2016) found a mix of positive and negative sensitivities across the 21$^{st}$ century, depending on scenario. Sulfate concentrations declined strongly by the 2090s in all future model scenarios, contrary to what our results imply. Our results suggest that these chemistry models may

underestimate the impact of future climate change on U.S. $PM_{2.5}$ air quality.

Using GEOS-Chem, we further explore the sensitivity of monthly mean $PM_{2.5}$ to temperature in the summertime. We regress the simulated monthly mean concentrations of key $PM_{2.5}$ components – sulfate, ammonium, organic aerosols (OA)

and BC – onto temperature over the 2004-2012 summers. In the observations, the positive slopes in sulfate-temperature and OA-temperature clearly drive the positive $PM_{2.5}$-temperature slopes (Figure 6a, 6c and 6e). In GEOS-Chem, the OA-temperature slopes match those in the observations (Figure 6e-f), but the modeled sulfate-temperature slopes exhibit negative values in the South (Figure 6d), contrary to observations (Figure 6c). For other $PM_{2.5}$ species, the slopes with temperature are relatively weak, with minimal contributions to the total $PM_{2.5}$-temperature slopes in both observations and GEOS-Chem

(Figure 6g-j). The observed ammonium-temperature slopes are weakly positive over the East, but are positive in the Northeast and negative in the Southeast in GEOS-Chem, in a spatial pattern similar to that of modeled sulfate-temperature (Figure 6g-h). The nitrate-temperature slopes are negligible in AQS observations but weakly negative over the East in GEOS-Chem (Figure 6i-j). For both ammonium and nitrate, GEOS-Chem underestimates the dependence on temperature, indicating that the model likely has difficulty in simulating the competition between increased emission and faster evaporation at higher temperatures. In any event, Figure 6 makes clear that the underestimate of $PM_{2.5}$-temperature slopes in GEOS-Chem is mainly caused by the underestimate in sulfate-temperature slopes.

We next search for the reasons of the underestimate in sulfate-temperature slopes in GEOS-Chem. Three important pathways for sulfate oxidation chemistry exist: gas-phase oxidation by OH and aqueous-phase oxidation by either $H_2O_2$ or $O_3$ (Jacob 1999). Total sulfate production rate is much greater in the eastern United States due to abundant anthropogenic emissions there. The relative importance of these three pathways varies by region: in summer, aqueous-phase oxidation by $H_2O_2$ is most important in the East, while gas-phase oxidation by OH dominates in the West. We calculate the monthly total sulfate production rates (kg month$^{-1}$ grid$^{-1}$) in each pathway and then regress them onto the monthly temperature in summer. As demonstrated by Figure 7a, as temperature increases, OH oxidation rates in GEOS-Chem vary little. In contrast, modeled $H_2O_2$ oxidation rates decrease rapidly with temperature in the South and increase significantly in the Northeast (Figure 7b), displaying a similar spatial pattern as the sulfate-temperature slopes in Figure 6d. Modeled $O_3$ oxidation rates also decrease with temperature in the South (Figure 7c), but with slopes much smaller than those of the $H_2O_2$ oxidation rates. Given that atmospheric $SO_2$, $H_2O_2$, and $O_3$ concentrations all increase with temperature in GEOS-Chem (not shown), our results suggest that the relationship of cloud fraction and temperature may not be well parameterized in GEOS-5, the earth system model which provides the meteorology driving GEOS-Chem. In GEOS-5, cloud fraction is not assimilated from observations but is calculated online as a prognostic variable (Suarez et al., 2008).

As a check on our hypothesis, we compare the sensitivity of cloud fraction to temperature in GEOS-5 with that in the ISCCP-D2like D2 product from CERES satellite observations. We focus on cloud fraction in the lower troposphere (> 680 hPa), as surface sulfate $PM_{2.5}$ is likely most responsive to oxidation in this part of the atmosphere Because no reliable observations of nighttime cloud fraction exist, we focus on daytime measurements. On average, increased cloud fraction is associated with cooler surface air temperatures, but the relationship between cloud fraction and temperature can also have a strong seasonal cycle and vary by region (Groisman et al., 2000; Sun et al., 2000). Figure 8 shows the slopes of monthly mean cloud fraction (>680 hPa) and surface temperature in summer from 2004 to 2012 over the Southeast in daytime. The satellite observations yield relatively weak slopes (±0.01 K$^{-1}$), but GEOS-5 displays strongly negative slopes (~-0.04 K$^{-1}$). This result suggests that cloud fraction in GEOS-5 is too sensitive to temperature, which in turn makes aqueous-phase oxidation rates decrease too rapidly as temperature increases in the South and leads to negative sulfate-temperature slopes.

With regard to the ACCMIP results, understanding the failure of these models to capture the observed slopes of monthly mean total $PM_{2.5}$ and temperature is beyond the scope of this paper. Key diagnostics, such as the production rates of sulfate through different oxidation pathways, are not available.

## 6 Discussion and Conclusions

In this study, we use a statistical model to investigate the effect of 2000-2050 climate change on fine particulate matter ($PM_{2.5}$) air quality across the contiguous United States. To our knowledge, this study represents the first time that the influences of both local meteorology and synoptic circulations are considered in projecting future changes in $PM_{2.5}$ air quality. We have developed a new method to characterize $PM_{2.5}$-related circulation patterns, using Singular Value Decomposition (SVD) of the spatial correlations between $PM_{2.5}$ and meteorological variables across the surrounding region (~1,000 km). Our regression model uses both these synoptic-scale relationships and relationships of $PM_{2.5}$ with local meteorology. Use of SVD increases the explained variability in 1999-2013 monthly $PM_{2.5}$ across the United States from 34%, when only local meteorology is considered, to 43%.

To estimate the impacts of climate change on future $PM_{2.5}$ concentrations from 2000-2019 to 2050-2069, we apply our regression model to the CMIP5 future meteorological projections from an ensemble of 19 GCMs under the RCP4.5 scenario. The average change in $PM_{2.5}$ across models provides a robust estimate of the climate impact on U.S. $PM_{2.5}$, and the spread of projected changes allows us to determine the statistical significance of the average. Assuming that anthropogenic emissions remain at present-day levels, we project an increase of ~0.4-1.4 μg m$^{-3}$ in annual mean $PM_{2.5}$ in the eastern US and a decrease of 0.3-1.2 μg m$^{-3}$ in the Intermountain West. Mean summer $PM_{2.5}$ increases as much as 2-3 μg m$^{-3}$ in the eastern United States due to faster oxidation and greater biogenic emissions. Mean winter $PM_{2.5}$ decreases by 0.3-3 μg m$^{-3}$ over most regions in United States probably due to the volatilization of ammonium nitrate.

Previous model simulations show no consistent sign of the future $PM_{2.5}$ changes under a warmer climate (Jacob and Winner, 2009; Fiore et al., 2015), and the magnitudes of these changes are much smaller than this study. We examine the ability of four different atmospheric chemistry models to simulate the observed relationship between $PM_{2.5}$ and temperature. Results show that these models underestimate or even fail to capture the observed positive relationship between monthly mean $PM_{2.5}$ and temperature in the eastern United States in summer, implying they may also underestimate future changes in $PM_{2.5}$ under a warmer climate regime. By comparing with in-situ observations, we find that the discrepancies of monthly mean $PM_{2.5}$-temperature slopes in GEOS-Chem are mainly caused by the underestimate of sulfate-temperature slopes, which in turn appears related to deficiencies in the parameterization of cloud processes in GEOS-5, the earth system model that provides assimilated meteorology for GEOS-Chem. The 2004-2012 slopes of monthly mean cloud fraction (> 680 hPa) and surface temperature are relatively weak (±0.01 K$^{-1}$) in satellite observations but strongly negative (~-0.04 K$^{-1}$) in GEOS-5 over the

Southeast in daytime. This result suggests that cloud fraction, a prognostic variable in GEOS-5, is too sensitive to temperature and that the rate of aqueous-phase $H_2O_2$ oxidation in GEOS-Chem decreases too rapidly with increasing temperature. This hypothesis would explain the negative sulfate-temperature slopes in GEOS-Chem in the South, in contrast to the positive slopes in observations. Other chemistry models may have similar problems in cloud fraction or other

variables important to $PM_{2.5}$ production or loss.

CTMs and CCMs are frequently applied to predict future air quality. Our work underscores the importance of evaluating the skill of such models to simulate long-term relationships between $PM_{2.5}$ and temperature and perhaps other variables. Without such evaluations, the credibility of future model projections of $PM_{2.5}$ is not clear. Drawbacks of this study include

its assumption of constant anthropogenic emissions and its dependence on a relative short history (~15 years) of $PM_{2.5}$ observations. We also do not explicitly consider the role of interannual variability in the climate system and how that might influence our results (Deser et al., 2013). Within these limitations, this study provides an up-to-date, observationally-based prediction of future $PM_{2.5}$ with relevance for air quality management. It also demonstrates the utility of a computationally efficient model whose projections of the climate penalty on air quality can be readily compared to those from more

traditional dynamic models.

**Data availability**

All datasets used in this study are publically accessible.

**Author contributions**

L. Shen and L. Mickley designed the experiments. L. Shen developed the model code and performed most experiments. L. Murray performed the GISS-ModelE2 simulations. L. Shen prepared the manuscript with contributions from all co-authors.

**Competing interests**

The authors declare that they have no conflict of interest.

**Acknowledgments**

We thank for the guidance in cloud fraction analysis from Hongyu Liu in National Institute of Aerospace at NASA Langley Research Center. This work was supported by the National Aeronautics and Space Administration (NASA Air Quality Applied Sciences Team and NASA-MAP NNX13AO08G), the National Institute of Environmental Health Sciences (NIH R21ES022585), and the Environmental Protection Agency (EPA-83575501-0). This publication was developed under Assistance Agreement 83575501-0 awarded by the U.S. Environmental Protection Agency. It has not been formally reviewed by EPA. The views expressed in this document are solely those of the authors and do not necessarily reflect those of the Agency. EPA does not endorse any products or commercial services mentioned in this publication.

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

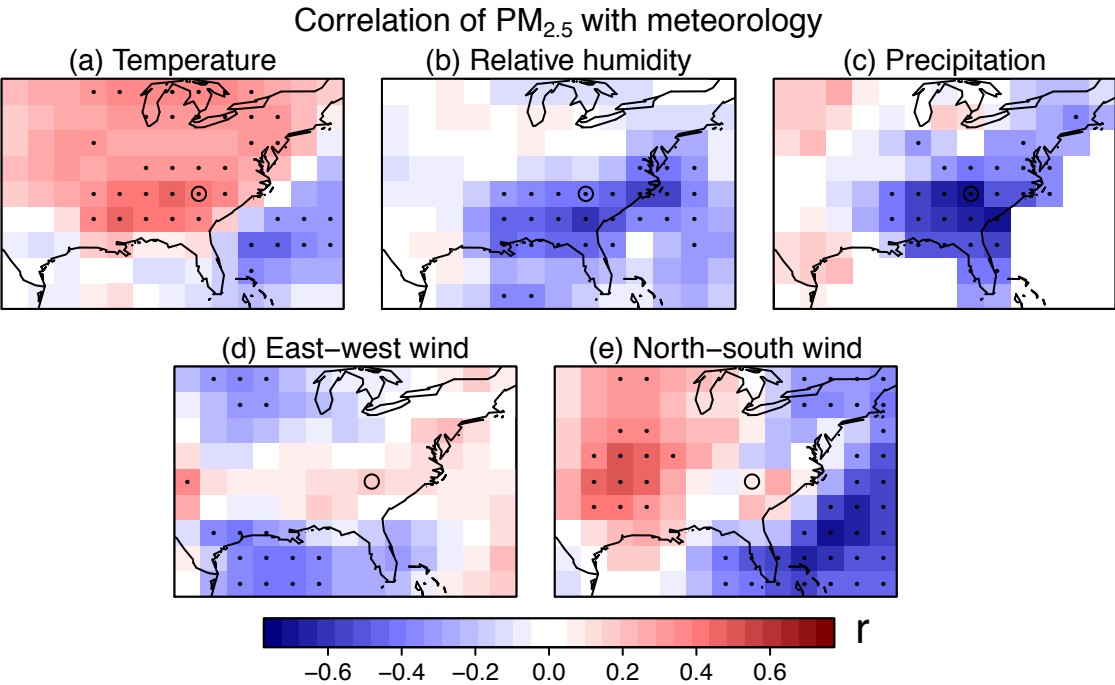

5  **Figure 1**. Example of observed correlations of monthly mean $PM_{2.5}$ in one grid box with surrounding meteorology in the southeast United States from 1999-2013.  Panels show correlations of May-June-July monthly $PM_{2.5}$ concentrations from EPA-AQS observations in the 2.5°×2.5° grid box centered at 82.5°W, 32.5°N (black circle) with different meteorological variables from NCEP Reanalysis1, including (a) surface air temperature, (b) relative humidity, (c) total precipitation, (d) east-west wind speed and (e) north-south wind speed. Grid boxes with significant correlation with $p < 0.05$ level are stippled.

10 All data are detrended by subtracting the 5-year moving average from the monthly values.

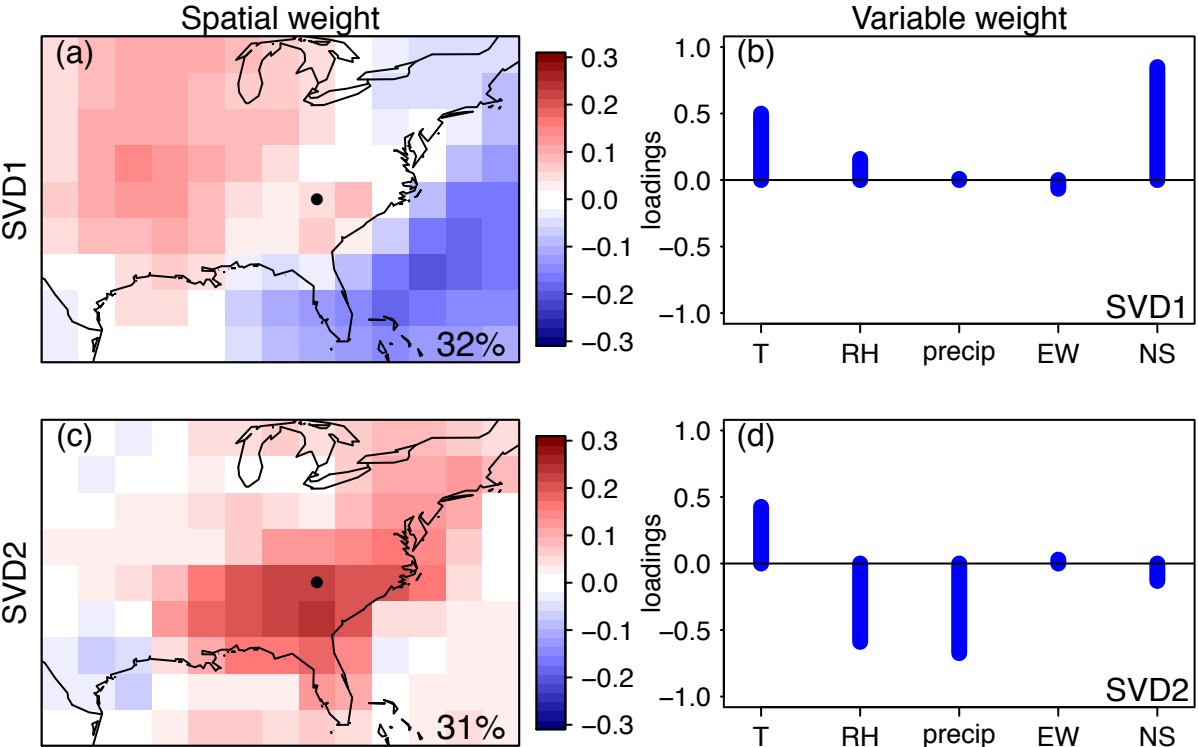

**Figure 2**. (a, c) Spatial and (b, d) variable weights of the (a, b) first and (c, d) second Singular Value Decomposition (SVD) modes describing the spatial correlations of May-June-July $PM_{2.5}$ anomalies in one grid box in the Southeast from 1999-2013 and five different meteorological variables: temperature (T), relative humidity (RH), precipitation (precip), and east-west and north-south wind speed (EW-wind and NS-wind). The explained variance by each SVD mode is shown inset. See Section 3 for more details.

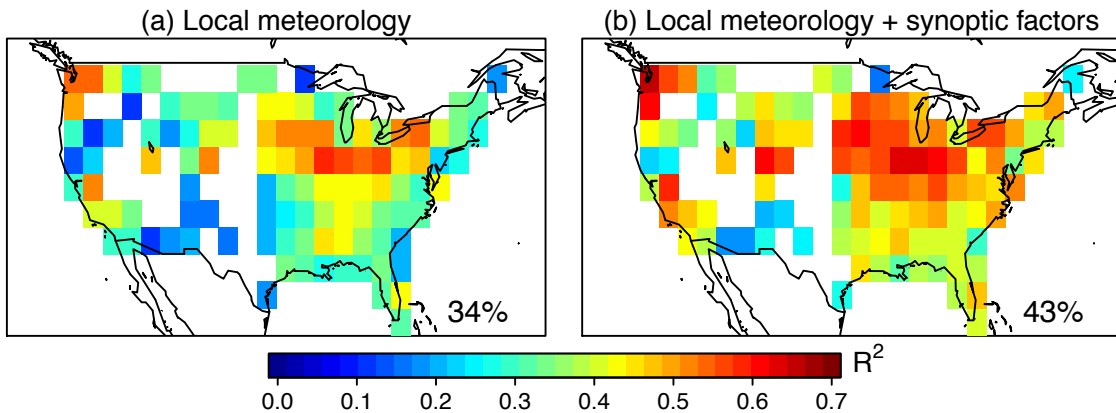

**Figure 3**. Cross-validated coefficients of determination ($R^2$) between observed and predicted 1999-2013 monthly $PM_{2.5}$ across the United States, calculated with (a) local meteorological variables and (b) both local meteorology and patterns of synoptic circulation. Spatially averaged coefficients of determination are shown inset.

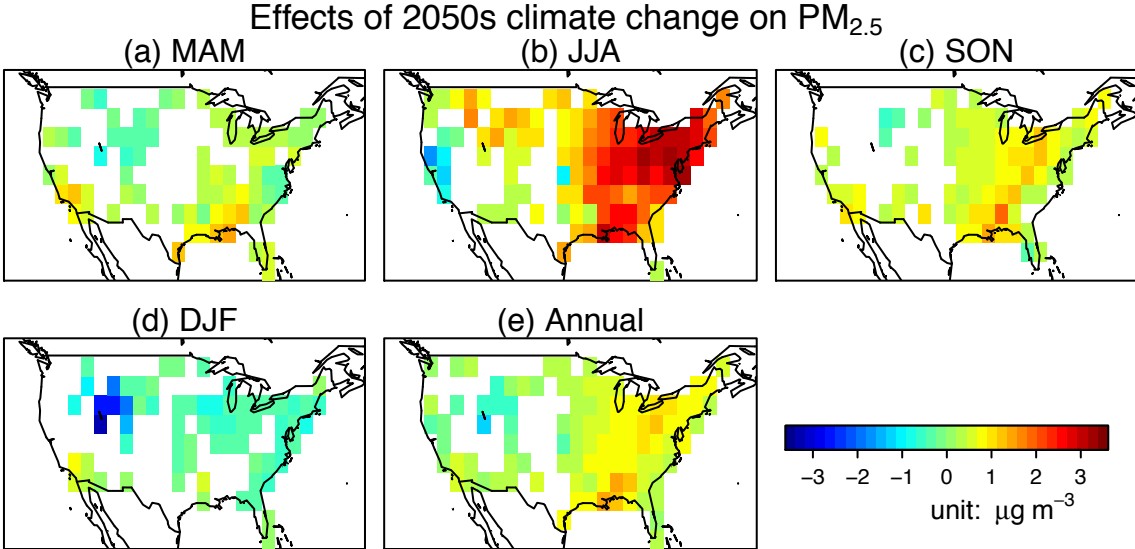

**Figure 4**. Effects of climate change from 2000-2019 to 2050-2069 on (a-d) seasonal and (e) annual mean PM$_{2.5}$ concentrations, calculated with observed relationships of PM$_{2.5}$ and meteorology and with meteorology projected by an ensemble of 19 CMIP5 models. The panels show the mean change in surface PM$_{2.5}$, averaged across the projections. White areas refer to the regions with no PM$_{2.5}$ observations or where fewer than 14 models yield the same sign of change.

## JJA slopes of PM$_{2.5}$ and temperature

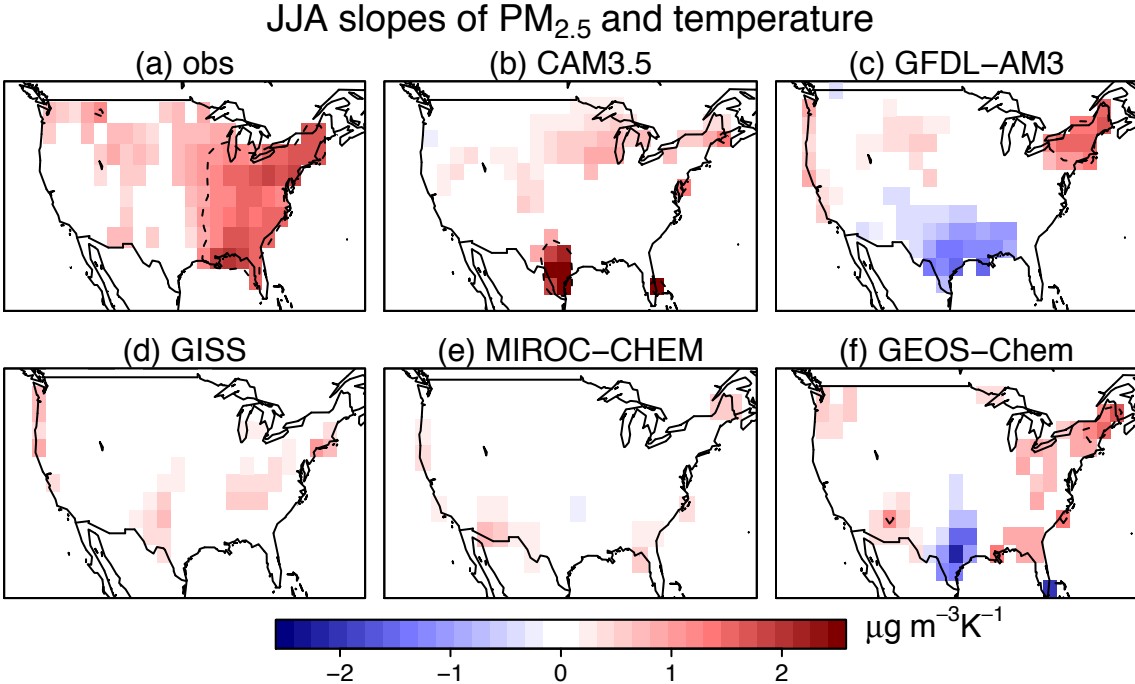

**Figure 5**. The slopes of detrended, monthly mean PM$_{2.5}$ versus temperature for summer months (June-July-August) in (a) observations and (b-f) different chemistry models. The timeframes shown in the panel are as follows: (a) 2004-2012, (b) 2002-2009, (c) 2001-2010, (d) 1995-2005, (e) 2000-2010 and (f) 2004-2012. Results in panels (b-e) are taken from ACCMIP [Lamarque et al., 2010], and use an updated GISS simulation (d) relative to their ACCMIP contributions (See text for more details). The dashed contour line in some panels denotes a slope of +1 $\mu$g m$^{-3}$ K$^{-1}$. White areas indicate either missing data or grid boxes where the slope is not significant at the 0.05 level.

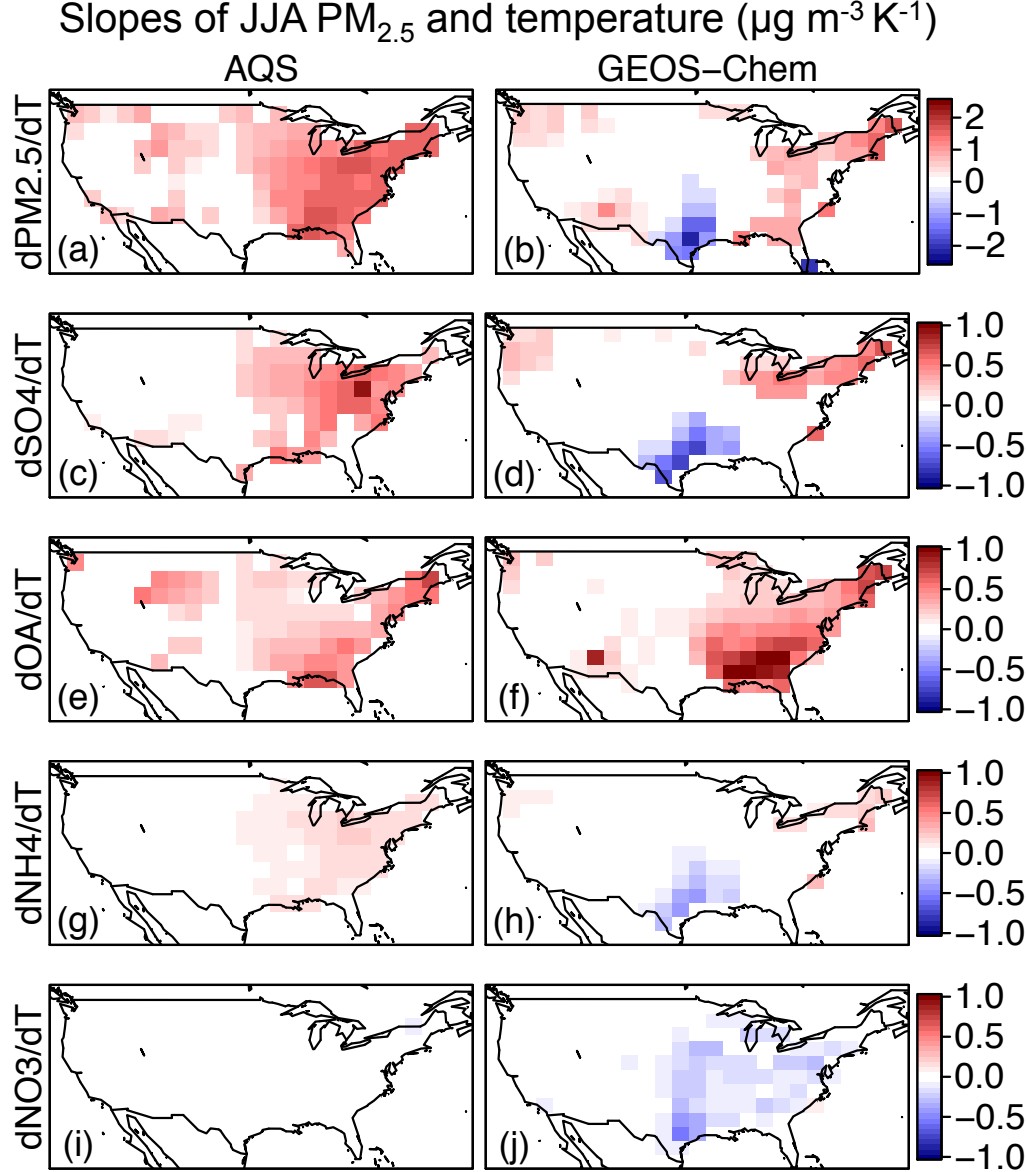

**Figure 6**. The slopes of detrended (a-b) monthly mean PM$_{2.5}$ and (c-j) different PM$_{2.5}$ components with surface air temperature for 2004-2012 summer months. Left column shows slopes from AQS observations, and right column shows results from GEOS-Chem. Organic aerosol (OA) in Panel e is inferred from the measured organic carbon (OC) component using an OA/OC mass ratio of 1.8 (Canagaratna et al., 2015). Panels a and b are the same as Figures 5a and 5f. White areas indicate either missing data or grid boxes where the slope is not significant at the 0.05 level.

**Figure 7**. Slopes of monthly mean sulfate production with surface air temperature for 2004-2012 summer months, as calculated by GEOS-Chem. The panels show slopes from three different production pathways: (a) gas-phase oxidation by OH and aqueous-phase oxidation by (b) $H_2O_2$ and (c) $O_3$. See Section 5 for more details. White areas indicate either missing data or grid boxes where the slope is not significant at the 0.05 level.

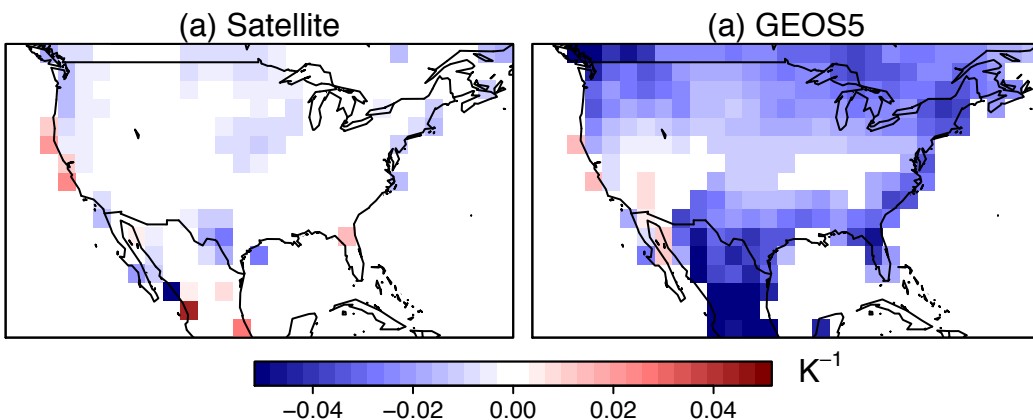

**Figure 8**. Daytime slopes of monthly mean cloud fractions in the lower troposphere (> 680 hPa) versus surface air temperature over land for June-July-August from 2004 to 2012 in (a) the merged ISCCP-D2like products from CERES and (b) GEOS-5 meteorology. White areas indicate the slope is not significant at the 0.05 level.