# Peer review of "Influence of 2000-2050 climate change on particulate matter in the United States: Results from a new statistical model"

_Atmospheric Chemistry and Physics, 2016_

## Referee Comment (RC1) · Anonymous Referee #1 · 27 Nov 2016

This study describes a new statistical approach to characterizing both local and synoptic meteorological impacts on PM2.5 air quality. The authors develop the statistical relationships based on over a decade of PM2.5 observations over the United States, and then apply these to the ACCMIP models and the GEOS-Chem model to predict the influence of changing climate on PM2.5 concentrations in 2050. They identify the strongest relationship between PM2.5 and temperature and characterize how this is represented by 4 models. They explore the specific response of the GEOS-Chem simulated PM2.5 to temperature in greater detail.

This is a nice study, with a new approach to exploring the meteorological processes controlling air quality. There are a few major points that the authors should address

prior to publication, the substance of these comments is to expand upon the discussion of the analysis to improve the clarity of the paper. I detail these below, followed by more minor comments.

1. I felt the discussion of the results was a bit superficial. Particularly with regards to the application of the SVD+local statistical relationships to the ACCMIP models. How did the model predictions vary? Were they robust in all regions? The manuscript suggests that the uncertainty in the estimate of the climate impact on PM2.5 can be characterized by using this suite of models (page 12, line 17), but they do not provide estimates of uncertainty or significance. The results in Figure 6 could also use more discussion (page 12, lines 2-3 is a little oversimplistic); the patterns look similar between GFDL and GEOS-Chem, though they are using very different meteorology (whereas GISS is driven by similar meteorology to GEOS-Chem). Perhaps the authors could comment on how the T and PM2.5 patterns compare between the models and obs? If the authors could add a little more discussion of their results, the paper would be much improved.

2. I also found that much (if not all) the supplementary material should be included in the main text. Many of the figures in supplementary are discussed extensively in the main text, and therefore should be more easily accessible.

3. The authors should justify their choice of meteorological variables. Why (only) surface T, RH, precipitation, and E-W & N-S wind speed as predictors?

4. How important is non-stationarity of emissions to the results? There are two aspects here: the changes in anthropogenic emissions (even removing a 5 year moving average of PM2.5 will not eliminate long-term changes in anthropogenic emissions over the 14 year record. Are the statistical relationships similar if the authors use only the early or only the later part of the record?). Secondly: is the 14 year record sufficient for significant T-driven changes in BVOC to impact OA? I assume that this is what the authors are suggesting on page 9 line 13 as the reason for the projected increase in summertime PM2.5 in the eastern US (if not, please clarify in the text), however, it's

not clear that this relationship would be identifiable in the statistical analysis. Please discuss.

5. The authors did not discuss the impact of covariance on their analysis. The meteorological variables are not all statistically independent. How well correlated are the SVD patterns with the local meteorology? How does this impact the results?

Additional Comments

1. Title: "Strong influence" seems overstated. Strong compared to what? Compared to changes in emissions, these climate-driven responses are not large changes in PM2.5. I suggest that the authors remove the word "Strong"

2. Page 1, Line 9: "we bypass many of the uncertainties inherent in chemistry-climate models", seems a bit overstated. The authors have developed a statistical approach which is complementary to chemistry-climate model predictions, but not without its own limitations. I suggest that the language be softened.

3. Page 2, Line 4: I suggest that the authors cite the relevant epidemiological literature for these statements rather than the application studies of Lelieveld et al.

4. Page 2, Line 12: "to more robustly quantify" is a very strong claim which is impossible to substantiate. I suggest that the authors soften their language.

5. Page 3, Line 10 & 12: "In contrast" and "inconsistencies" suggests that Day et al. (2015) and Val Martin et al. (2015) disagree, but in fact the results discussed are for different time periods (summer vs annual) and different scenarios. Therefore they are not necessarily in disagreement. Either compare similar results, or modify language.

6. Page 3, Line 24: define T

7. Page 3, Line 26: what does "period T" mean?

8. Page 5, line 10-20: what biomass burning emissions are used in the model. Do they vary year-to-year? If so, how might this impact the analysis? More generally, it would

be useful to comment on the role of fire emissions (as a possible feedback from climate change) in this analysis.

9. Page 5, line 28-29: This last sentence seems out of place as the suggested analysis does not follow. Please indicate in which section this analysis will be discussed in the paper.

10. Page 6, line 21: "making clear" seems a bit strong. The results are suggestive of a regional climate influence. They may also be indicative of a relatively homogeneous region.

11. Section 3: the time horizon for the analysis is not always clear. It would be helpful if you could clarify the time resolution of the analysis (monthly, as I understand it?), as you present both seasonal and annual averages in the results.

12. Page 7, line 8: identify which dimension corresponds with which variable in matrix A

13. Page 7, line 16: I believe that the authors mean to refer to Figure 1e, not 1d

14. Page 7, line 19: typo? "negative" looks like positive anomalies in the figure? Also these are only seen in Figure 1a (not 1a-1c as indicated in the text).

15. Figure 1 caption indicates that the analysis was for summer. Figure 2 caption does not indicate the time horizon. These should be consistent for the authors to compare them. Please update Figure 2 caption and ensure consistency.

16. Line 11: how were the results from the 17 models combined in Figure 4?

17. Page 9, line 14: "driven by". Be careful with the language, this is speculation not attribution.

18. Page 10, lines 4-5: May be worth noting that not that many studies have investigated the climate impact on PM2.5 (compared to say O3) and that PM2.5 consists of many different chemical species, so a more complex system to understand the re-

sponse.

---

## Referee Comment (RC2) · Anonymous Referee #2 · 5 Dec 2016

I believe the study presents several analyses investigating projections of climate change impacts on PM2.5 pollution that provide valuable insights to the air quality modeling community. The manuscript is well-written and clear. I appreciate the authors' effort to undertake a study that includes several layers of research: developing and describing a new statistical regression model, applying the model to the projections of a multi-model GCM ensemble, using these results to guide an investigation into PM2.5 projections from CCMs, and using a CTM to identify factors contributing to the inconsistencies in simulations of PM2.5 impacts. As it stands, the study presents several useful findings that make it worthy of publication. However, by addressing different research needs and different modeling approaches (including a statistical regression

model, 17 different GCMs, 4 different CCMs, and 1 CTM), the analysis of results for each research question being investigated is at times limited. I encourage the authors' to go deeper in their discussion. I would also persuade the authors to further investigate the major findings of their work individually in follow-up research. Some specific comments are included below.

- I felt there is some disconnection between different aspects of the study as it moves from the regression model to GEOS-Chem. The study could be broken down into separate analyses: (1) a PM2.5/meteorology linear regression model; (2) projection of PM2.5 climate impact from the CMIP5 GCM ensemble; (3) PM2.5/temperature relation in 4 ACCMIP CCMs; (4) GEOS-Chem sensitivity of PM2.5 to temperature. The connection between (1) and (2) is evident, while the connection between subsequent sections is not as clear. In moving from sections 4 to 5, the manuscript goes from statistical inference of PM2.5 changes from 20-yr present/midcentury simulations with 17 GCMs, to atmospheric chemistry simulations from 4 CCMs covering a different 15-yr present period and conditions, to a CTM simulations for a different 9 yr period. Is there truly a clear connection between these different types of models and the nature of these simulations, other than saying that the temp-PM25 relationship is important? The scope of the study limits the depth with which each finding is examined.

- One topic I would encourage the authors to discuss further in their manuscript is the impacts of 2050 climate derived from the CMIP5 ensemble and the regression model. Only ensemble-mean results are presented. I would be very interested in seeing the differences in the projections from individual GCMs. Additionally, I would like to see the interannual variability across individual years in the 20-yr penalty estimates. How robust is the study's finding of an anthropogenic-induced climate change impact by 2050 under a stabilization scenario, given that natural variability has been shown to significantly influence U.S. temperature and precipitation projections on timescales as long as 50 years (Deser et al., 2014; doi: doi:10.1175/JCLI-D-13-00451.1.)? There is a great opportunity to explore climate-related uncertainty in PM2.5 projections at much

greater depth within this data.

- I suggest combining figures 4 and 5 and showing additional details of the penalty projections within the manuscript.

- One analysis that is absent but would greatly benefit the study is a comparison of a climate penalty projection generated by the regression model to that generated with a CTM for the same GCM meteorological fields. Comparing the midcentury climate penalty estimated with the regression model, to a projection generated by driving a CTM with the same weather fields (e.g. a GEOS-Chem simulation driven by the present/future met fields from one of the CMIP5 models) would provide great insight into the potential to replace computationally expensive CTMs with a statistical model, and limitations associated with either approach. I encourage the authors to undertake this analysis in future work.

- The penalty projections generated using the regression model assume that observed relationships between PM2.5 and meteorology remain valid at midcentury under significantly different meteorological conditions and emissions levels. Is this an adequate assumption? An interesting analysis would be to compare the penalty projections of regression models generated under different levels of emissions within the 15-yr observational record.

- The study explores PM2.5 response to surface temp. in 4 CCMs, and then further investigates the dependency in GEOS-Chem. Given the differences between GEOS-Chem (a CTM) and CCMs, what insights from the GEOS-Chem analysis may useful to identify the causes for the discrepancies between observed and simulated sensitivities in CCMs? Would the authors expect to see any similarities?

- PM2.5 concentrations and meteorology could be mapped on a finer resolution grid, and CMIP5 fields interpolated onto that grid. Would there be a benefit or significant change if the statistical regression model were built at higher resolution, rather than the coarse 2.5°x2.5°?

- When listing the range of reported projections for climate change impacts on PM2.5 (e.g. pg. 3, line 7), I recommend using the updated range from the reviews published by Fiore et al. (doi: 10.1080/10962247.2015.1040526.)
* * *

---

## Author Comment (AC1) · 16 Feb 2017

The comment was uploaded in the form of a supplement:
http://www.atmos-chem-phys-discuss.net/acp-2016-954/acp-2016-954-AC1-supplement.zip

---

## Author Comment (AC2) · 16 Feb 2017

The comment was uploaded in the form of a supplement:
http://www.atmos-chem-phys-discuss.net/acp-2016-954/acp-2016-954-AC2-supplement.zip

---

## Author Response (AR2)

**Response to referee comments on "Strong influence of 2000-2050 climate change on particulate matter in the United States: Results from a new statistical model"**

We thank the referees for their careful reading of the manuscript and the valuable comments. This document is organized as follows: the Referee's comments are in *italic*, our responses are in plain text, and all the revisions in the manuscript are shown in blue. **Boldface blue** text denotes text written in direct response to the Referee's comments. The line numbers in this document refer to the updated manuscript.

**Reviewer 1**

*This study describes a new statistical approach to characterizing both local and synoptic meteorological impacts on $PM_{2.5}$ air quality. The authors develop the statistical relationships based on over a decade of PM2.5 observations over the United States, and then apply these to the ACCMIP models and the GEOS-Chem model to predict the influence of changing climate on PM2.5 concentrations in 2050. They identify the strongest relationship between PM2.5 and temperature and characterize how this is represented by 4 models. They explore the specific response of the GEOS-Chem simulated PM2.5 to temperature in greater detail.*

*This is a nice study, with a new approach to exploring the meteorological processes controlling air quality. There are a few major points that the authors should address prior to publication; the substance of these comments is to expand upon the discussion of the analysis to improve the clarity of the paper. I detail these below, followed by more minor comments.*

**Response**: Thanks for raising so many good points. This feedback has significantly improved the manuscript. We also wish to draw the reviewer's attention to the fact that we have increased the number of CMIP5 models used in this study from 17 to 19. All results are now based on this ensemble of 19 models; this change has very little influence on our previous results. For a full list of these models, please see Table S1. We have also replaced mass of organic carbon (OC) with the inferred mass of organic aerosol (OA) in Figures 6 and S13. OC is the measured carbon component of OA.

*1. (a) I felt the discussion of the results was a bit superficial. (b) Particularly with regards to the application of the SVD+local statistical relationships to the ACCMIP models. How did the model predictions vary? Were they robust in all regions? (c) The manuscript suggests that the uncertainty in the estimate of the climate impact on PM2.5 can be characterized by using this suite of models (page 12, line 17), but they do not provide estimates of uncertainty or significance. (d)The results in Figure 6 could also use more discussion (page 12, lines 2-3 is a little oversimplistic); the patterns look similar between GFDL and GEOS-Chem, though they are using very different meteorology (whereas GISS is driven by similar meteorology to GEOS-Chem). Perhaps the authors could comment on how the T and PM2.5 patterns compare between the models and obs? If the authors could add a little more discussion of their results, the paper would be much improved.*

**Response**: Since this is a long question, we decompose it into four parts and answer each part one by one.

Part (a)
We have added more discussion of our results, including discussion of uncertainty in our study, the potential effect of increasing wildfires in the future climate, other recent studies and how they compare with ours, the impacts of changing anthropogenic emissions on $PM_{2.5}$ concentrations, and Figures S5-S11 in the supplement. Please refer to the highlighted blue text in Section 4-5. Much of the new content was written in response to the reviewer's comments, as detailed below.

Part (b)

We have added a new Figure in the Supplement that displays the model predictions in each season. The main text refers to the new Figure as follows.

Page 8, Line 32. We also find that the cross-validated values of $R^2$, calculated from both local meteorology and patterns of synoptic circulation and averaged across the United States, are 35% in spring, 44% in summer, 42% in autumn and 43% in winter (Figure S1).

[Figure]

**Figure S1.** Cross-validated coefficients of determination ($R^2$) between observed and predicted 1999-2013 monthly $PM_{2.5}$ in different seasons across the United States, calculated with both local meteorology and patterns of synoptic circulation. Spatially averaged coefficients of determination are shown inset.

Part (c)

We have made the following changes to show both the significance and uncertainty of the changes in $PM_{2.5}$ concentrations among the 19 CMIP5 models.

First, to estimate the significance, we show only those changes for which more than 14 models yield the same sign of change. We combine the old Figure 4 and 5 into one single figure, as suggested by the second reviewer.

[Figure]

**Figure 4**. Effects of climate change from 2000-2019 to 2050-2069 on (a-d) seasonal and (e) annual mean PM$_{2.5}$ concentrations, calculated with observed relationships of PM$_{2.5}$ and meteorology and with meteorology projected by an ensemble of 19 CMIP5 models. The panels show the mean change in surface PM$_{2.5}$, averaged across the projections. **White areas refer to the regions with no PM$_{2.5}$ observations or where fewer than 14 models yield the same sign of change.**

Second, we add two figures to show the 90[th] and 10[th] percentile changes of PM$_{2.5}$ across the projections among the 19 CMIP5 models. We also discuss about these two figures in the main text.

Page 10, line 9-14. To more rigorously characterize this uncertainty, we calculate the 90[th] and 10[th] percentile changes in PM$_{2.5}$ concentrations as calculated from the 19 CMIP5 models (Figure S6-S7). In the summertime, the 90[th] percentile changes of PM$_{2.5}$ can be greater than 3 μg/m$^3$ across most of the eastern United States (Figure S6b), but the 10[th] percentile changes are only 0.5-1.5 μg/m$^3$ (Figure S7b). These discrepancies underscore the importance of using an ensemble of climate models to project future PM$_{2.5}$ concentrations. Such an approach allows us to identify robust results across models, quantify uncertainty, and diagnose model outliers.

[Figure]

**Figure S6**. Same as Figure 4, but for the 90th percentile changes of $PM_{2.5}$ concentrations, calculated with meteorology projected by the ensemble of 19 CMIP5 models.

[Figure]

**Figure S7**. Same as Figures 4 and S7, but for the 10th percentile changes of $PM_{2.5}$ concentrations, calculated with meteorology projected by the ensemble of 19 CMIP5 models.

Part (d)
We now give detailed description of the slopes of $PM_{2.5}$ and temperature generated by the ACCMIP models.

Page 12, line 17-23. For example, CAM3.5 shows significant positive slopes in Texas, the Midwest, and Northeast (Figure 5b). GFDL-AM3 displays a bimodal structure, with positive slopes in the Northeast but negative slopes in the South (Figure 5c). The GISS-ModelE2 shows

slight positive slopes over parts of the East (Figure 5d). The slopes in MIROC-CHEM are nearly flat, indicating little sensitivity of the monthly mean $PM_{2.5}$ concentrations to temperature variability (Figure 5e). GEOS-Chem shows positive slopes over much of the eastern United States, but the magnitudes are much less than those observed (Figure 5f).

We are unable to provide more insights into the reasons for the failure of these models to capture the observed sensitivity of $PM_{2.5}$ and temperature. Key diagnostics, such as the production rates of sulfate through different oxidation pathways, are not available. We do, however, provide a very detailed analysis for the failure in GEOS-Chem, a model that we know well. We now clarified this issue in the manuscript.

 (We replace the sentence above with new text, shown below.)

Page 14, line 1-3. With regard to the ACCMIP results, understanding the failure of these models to capture the observed slopes of monthly mean total $PM_{2.5}$ and temperature is beyond the scope of this paper. Key diagnostics, such as the production rates of sulfate through different oxidation pathways, are not available.

*2. I also found that much (if not all) the supplementary material should be included in the main text. Many of the figures in supplementary are discussed extensively in the main text, and therefore should be more easily accessible.*

**Response**: Thank you. We have moved the old Figure S4 and S5 back to the main text.

*3. The authors should justify their choice of meteorological variables. Why (only) surface T, RH, precipitation, and E-W & N-S wind speed as predictors?*

**Response**: We have clarified this choice in the main text.
Page 4, Line 21-24. These variables have been used previously to predict $PM_{2.5}$ (e.g., Tai et al., 2010, 2012a, 2012b; Lecœur et al., 2014), and their variability is closely linked to that of synoptic patterns (e.g., Shen et al., 2015; Thishan Dharshana et al., 2010). These particular variables have also been validated in CMIP5 models (e.g., Sheffield et al., 2012).

*4. How important is non-stationarity of emissions to the results? There are two aspects here: the changes in anthropogenic emissions (even removing a 5 year moving average of PM2.5 will not eliminate long-term changes in anthropogenic emissions over the 14 year record. Are the statistical relationships similar if the authors use only the early or only the later part of the record?). Secondly: is the 14 year record sufficient for significant T-driven changes in BVOC to impact OA? I assume that this is what the authors are suggesting on page 9 line 13 as the reason for the projected increase in summertime $PM_{2.5}$ in the eastern US (if not, please clarify in the text), however, it's not clear that this relationship would be identifiable in the statistical analysis. Please discuss.*

**Response**: These are good questions.

First question.

We have added one figure in the supplement to show the slopes of JJA PM$_{2.5}$ as well as its components with temperature for 1999-2006 and 2007-2013. We also discuss the influence of changing emissions on the sensitivity of PM$_{2.5}$ to climate change.

Page 11, line 14-24. One weakness of this study is that when estimating the sensitivity of PM$_{2.5}$ to meteorological variables, we do not consider the impact of changing anthropogenic emissions on this sensitivity. Figure S13 compares the slopes of monthly mean PM$_{2.5}$ and its components with temperature for two time periods: 1999-2006 summers with high anthropogenic emissions and 1997-2013 summers with low anthropogenic emissions. Using the monthly data, we find that the changes of sensitivity of PM$_{2.5}$ to temperature vary across different locations and species. As the anthropogenic emissions decrease, the slopes of PM$_{2.5}$ and temperature decrease over the Great Plains and Midwest, but increase slightly in the south Atlantic States. Sulfate exhibits decreased sensitivity across the eastern United States, and OC shows no significant pattern of change. Reasons for such inconsistencies may be related to the shorter time periods and therefore less robust sensitivity. In this study, we have thus chosen not to investigate the influence of changing emissions on the sensitivity of PM$_{2.5}$ to climate change using this statistical model.

[Figure]

**Figure S13**. The slopes of detrended (a-b) monthly mean $PM_{2.5}$ and (c-j) different $PM_{2.5}$ components with surface air temperature for 1999-2013 summer months. Left column shows slopes for 1999-2006 with relatively high NOx emissions, and right column shows slopes for 2007-2013 with relatively low NOx emissions. Organic aerosol (OA) in Panel (e-f) is inferred from the measured organic carbon (OC) component using an OA/OC mass ratio of 1.8 (Canagaratna et al., 2015). White areas indicate either missing data or grid boxes where the slope is not significant at the 0.10 level. We note that the observation network has fewer sites in 1999 and 2000 than more recent years.

Second question.
Yes, the significant temperature-driven changes in BVOC can drive up OA in the future climate. This can be inferred from the observed relationship in of OC and temperature, as well as previous studies. Now we clarify this issue in the main text.

Page 9, line 26-29. $PM_{2.5}$ increases by ~2-3 μg/m$^3$ in summer in the eastern United States (Figure 4b), likely due to faster oxidation rates and more abundant organic aerosol in the warmer climate of the 2050s (e.g., Tai et al, 2010; Kelly et al, 2012; Gonzalez-Abraham et al., 2015). This can be also inferred from the positive sensitivity of sulfate and organic aerosols with temperatures from observations, which will be discussed in more details in Section 5.

Page 12, line 12. All $PM_{2.5}$ and temperature values have been detrended, as described above, so that the slopes reflect only the $PM_{2.5}$ response to the interannual variability in temperature.

Page 10, line 21-29. We also compare our results to those from recent studies using chemistry-climate models. Among the seven recent studies reviewed in Fiore et al (2015), only two of them projected a significant increase of $PM_{2.5}$ concentrations in summer over the eastern United States. Kelly et al. (2012) estimated an increase of 0.5-1.0 μg m$^{-3}$ in summertime $PM_{2.5}$ over much of the East from 2000 to 2050, mainly resulting from rapid increases in secondary organic aerosols from biogenic emissions. Gonzalez-Abraham et al. (2015) found that the effect of 2000-2050 climate change alone without changes in biogenic emissions can increase $PM_{2.5}$ concentrations by up to 1.0 μg m$^{-3}$ in the eastern United States, a combined effect of increasing sulfate and ammonium as well as decreasing nitrate. Consideration of the changes in biogenic emissions drives up this increase to 0.5-3 μg m$^{-3}$.

*5. The authors did not discuss the impact of covariance on their analysis. The meteorological variables are not all statistically independent. How well correlated are the SVD patterns with the local meteorology? How does this impact the results?*

**Response**: We thank the reviewer for pointing out this issue, which is a common challenge in statistical models. Yes, the SVD patterns are correlated with local meteorology, which can be inferred from the similarities between some of the correlation patterns in Figure 1 and the SVD patterns in Figure 2. In our method, however, we pick the best candidate variables from both the local meteorological variables as well as the synoptic patterns, using leave-one-out cross-validation, limiting the problem of multi-colinearity. To clarify this issue, we have added the following figure to the Supplement and added some discussion in the main text.

Page 9, line 2-5. To check the multi-colinearity among predictors in this model, we calculate the variance inflation factors (VIFs) for all variables in each gridbox and each month. Results in Figure S2 show that about 98.9% of the VIFs are less than 5, well below the threshold of 10 that defines significant multi-colinearity (Kutner et al., 2004).

[Figure]

**Figure S2.** The distribution of variance inflation factors (VIFs) of all variables in each gridbox and each month, calculated from the regression model using the best variable combination of both local meteorology and synoptic patterns.

Additional Comments
*1. Title: "Strong influence" seems overstated. Strong compared to what? Compared to changes in emissions, these climate-driven responses are not large changes in PM2.5. I suggest that the authors remove the word "Strong"*

**Response**: We have changed the title to "Influence of 2000-2050 climate change on particulate matter in the United States: Results from a new statistical model."

*2. Page 1, Line 9: "we bypass many of the uncertainties inherent in chemistry-climate models", seems a bit overstated. The authors have developed a statistical approach which is complementary to chemistry-climate model predictions, but not without its own limitations. I suggest that the language be softened.*

**Response**: The reviewer has a good point. We have slightly altered the wording.
Page 1, Line 9-10. By applying observed relationships of $PM_{2.5}$ and meteorology to the IPCC Coupled Model Intercomparision Project Phase 5 (CMIP5) archives, we bypass **some** of the uncertainties inherent in chemistry-climate models.

*3. Page 2, Line 4: I suggest that the authors cite the relevant epidemiological literature for these statements rather than the application studies of Lelieveld et al.*

**Response**: This is a good suggestion. Now we say:
Page 2, Line 3-4. Exposure to $PM_{2.5}$ can result in respiratory and cardiovascular disease, as well as premature mortality (e.g., Laden et al., 2006; Pellucchi et al., 2009; Brook et al., 2010).

*4. Page 2, Line 12: "to more robustly quantify" is a very strong claim which is impossible to substantiate. I suggest that the authors soften their language.*

**Response**: We have removed "more robustly" in that sentence.

Page 2, Line 12. In this study, we develop a new statistical model to  quantify the effect of 2000 to 2050 climate change on $PM_{2.5}$ air quality across the contiguous United States.

*5. Page 3, Line 10 & 12: "In contrast" and "inconsistencies" suggests that Day et al. (2015) and Val Martin et al. (2015) disagree, but in fact the results discussed are for different time periods (summer vs annual) and different scenarios. Therefore they are not necessarily in disagreement. Either compare similar results, or modify language.*

**Response**: We thank the reviewer for pointing this out. Now we compare Day et al. (2015) with Gonzalez-Abraham et al. (2015), both focused on the summer mean $PM_{2.5}$ concentrations.

Page 3, Line 10-14. More recently, val Martin et al. (2015) found that 2000-2050 climate change may decrease the annual mean $PM_{2.5}$ concentrations by 0-1 $\mu g\ m^{-3}$ in the eastern United States under the Representative Concentration pathway (RCP) 4.5 scenario of climate change. Day et al. (2015) determined that summer mean $PM_{2.5}$ increases by 21% in the Southeast but decreases 9% in the Northeast from 2000 to 2050 under the more greenhouse-gas intensive A2 scenario. In contrast, Gonzalez-Abraham et al. (2015) identified a 10-30% increase of summer mean $PM_{2.5}$ across the eastern United States by the 2050s.

*6. Page 3, Line 24: define T*

**Response**: Fixed.

Page 3, Line 26. ... with the average cyclone period *T*, defined as the inverse of the median frequency of the dominant meteorological mode…

*7. Page 3, Line 26: what does "period T" mean?*

**Response**: Now we say "average cyclone period T" instead of "period T"

*8. Page 5, line 10-20: what biomass burning emissions are used in the model. Do they vary year-to-year? If so, how might this impact the analysis? More generally, it would be useful to comment on the role of fire emissions (as a possible feedback from climate change) in this analysis.*

**Response**: We use biomass emission from GFED3 in GEOS-Chem. Now we clarify this in the text and add the discussion of fire emissions in the climate change analysis.

Page 5, line 27-28. We use monthly biomass burning emissions from Global Fire Emission Database (GFED, van der Werf et al., 2010).

Page 9, line 30-31. We also find an increase of ~0.8-1.5 $\mu g\ m^{-3}$ in the summer over the Intermountain West, partly driven by enhanced biomass burning in a warmer climate (e.g., Yue et al., 2013, 2015).

*9. Page 5, line 28-29: This last sentence seems out of place as the suggested analysis does not follow. Please indicate in which section this analysis will be discussed in the paper.*

**Response**: Fixed.
Page 6, Line 4-5. In Section 5, we validate the GEOS-5 cloud fraction in the lower troposphere against CERES satellite observations.

*10. Page 6, line 21: "making clear" seems a bit strong. The results are suggestive of a regional climate influence. They may also be indicative of a relatively homogeneous region.*

**Response**: Now we say "suggesting" instead of "making clear."
Page 6, line 29-30. Positive correlations extend across the whole Southeast, suggesting that $PM_{2.5}$ air quality in Georgia is affected by regional climate; the strongest correlations are located in Mississippi, ~500 km west of Georgia.

*11. Section 3: the time horizon for the analysis is not always clear. It would be helpful if you could clarify the time resolution of the analysis (monthly, as I understand it?), as you present both seasonal and annual averages in the results.*

**Response**: Yes, we predict the monthly $PM_{2.5}$ concentrations, but we show the seasonal and annual changes. We now clarify this issue in the text.
Page 8, line 22-24. Throughout this study, we predict monthly $PM_{2.5}$ concentrations using this regression model, but projected changes of $PM_{2.5}$ in the future climate will be displayed as seasonal and annual means.

*12. Page 7, line 8: identify which dimension corresponds with which variable in matrix A*

**Response**: Done.
Page 7, line 16. This step yields a 13×9×5 (longitude × latitude × variable) matrix which we call *A*.

*13. Page 7, line 16: I believe that the authors mean to refer to Figure 1e, not 1d*

**Response**: Fixed.

*14. Page 7, line 19: typo? "negative" looks like positive anomalies in the figure? Also these are only seen in Figure 1a (not 1a-1c as indicated in the text).*

**Response**: Sorry, this is a typo, now fixed.
Page 7, line 26-28. In the second SVD (SVD2) mode, the spatial weights (Figure 2c) show positive anomalies in the southeast United States, and this corresponds to the positive temperature anomalies in Figure 1a as well as negative relative humidity and precipitation anomalies in Figure 1b-c.

*15. Figure 1 caption indicates that the analysis was for summer. Figure 2 caption does not indicate the time horizon. These should be consistent for the authors to compare them. Please update Figure 2 caption and ensure consistency.*

**Response**: Fixed.

Figure 2. ...the spatial correlations of May-June-July $PM_{2.5}$ anomalies in one grid box in the Southeast from 1999-2013 and ...

*16. Line 11: how were the results from the 17 models combined in Figure 4?*

**Response**: Now we clarify this in the text.

Page 9, line 25-26. Figure 4a-d shows the response of the seasonal mean $PM_{2.5}$ concentrations to 2050s climate change across the United States, shown as the average of all projections from the CMIP5 models.

*17. Page 9, line 14: "driven by". Be careful with the language, this is speculation not attribution.*

**Response**: We now soften this language.

Page 9, line 32. In winter, future $PM_{2.5}$ decreases by 0.3-3 μg m$^{-3}$ across much of the United States (Figure 4d), **likely** driven by greater volatilization of ammonium nitrate at warmer temperatures (Dawson et al., 2007, 2009).

*18. Page 10, lines 4-5: May be worth noting that not that many studies have investigated the climate impact on PM2.5 (compared to say O3) and that PM2.5 consists of many different chemical species, so a more complex system to understand the response.*

**Response**: Since we don't know the exact number of the studies for ozone and $PM_{2.5}$, so we prefer to not make a comparison. We have made the following changes to address the second part of this comment.

Page 11, line 25-31. A key question is why previous model studies show no consistent sign in the in the change of future $PM_{2.5}$ relative to the present (Jacob and Winner, 2009). Such discrepancies no doubt arise in part because of differences in model projections of future climate or in model speciation of $PM_{2.5}$. In this section we investigate whether differences in model representation of the sensitivity of $PM_{2.5}$ to meteorological variability may also contribute to uncertainty in projections of future $PM_{2.5}$.

**Reviewer 2**

*I believe the study presents several analyses investigating projections of climate change impacts on PM$_{2.5}$ pollution that provide valuable insights to the air quality modeling community. The manuscript is well-written and clear. I appreciate the authors' effort to undertake a study that includes several layers of research: developing and describing a new statistical regression model, applying the model to the projections of a multi-model GCM ensemble, using these results to guide an investigation into PM$_{2.5}$ projections from CCMs, and using a CTM to identify factors contributing to the inconsistencies in simulations of PM$_{2.5}$ impacts. As it stands, the study presents several useful findings that make it worthy of publication. However, by addressing different research needs and different modeling approaches (including a statistical regression model, 17 different GCMs, 4 different CCMs, and 1 CTM), the analysis of results for each research question being investigated is at times limited. I encourage the authors' to go deeper in their discussion. I would also persuade the authors to further investigate the major findings of their work individually in follow-up research. Some specific comments are included below.*

**Response**: We thank the reviewer for raising so many good points. This feedback has significantly improved the manuscript. In particular, we have tried to deepen the discussion section in response to this reviewer's concerns.

We also to draw the reviewer's attention to the fact that we have increased the number of CMIP5 models used in this study from 17 to 19. All the results in the new manuscript will be based on the results from this ensemble of 19 models. For a full list of these models, please see Table S1. We have also replaced mass of organic carbon (OC) with the inferred mass of organic aerosol (OA) in Figures 6 and S13. OC is the measured carbon component of OA.

*- I felt there is some disconnection between different aspects of the study as it moves from the regression model to GEOS-Chem. The study could be broken down into separate analyses: (1) a PM2.5/meteorology linear regression model; (2) projection of PM2.5 climate impact from the CMIP5 GCM ensemble; (3) PM2.5/temperature relation in 4 ACCMIP CCMs; (4) GEOS-Chem sensitivity of PM2.5 to temperature. The connection between (1) and (2) is evident, while the connection between subsequent sections is not as clear. In moving from sections 4 to 5, the manuscript goes from statistical inference of PM2.5 changes from 20-yr present/midcentury simulations with 17 GCMs, to atmospheric chemistry simulations from 4 CCMs covering a different 15- yr present period and conditions, to a CTM simulations for a different 9 yr period. Is there truly a clear connection between these different types of models and the nature of these simulations, other than saying that the temp-PM25 relationship is important? The scope of the study limits the depth with which each finding is examined.*

**Response**: The reviewer raises valid concerns. In an effort to make the connections between sections more clear, we have rewritten much of the first paragraph of Section 5.

Page 11, line 29-31. A key question is why previous model studies show no consistent sign in the in the change of future PM$_{2.5}$ relative to the present (Jacob and Winner, 2009). Such discrepancies among models no doubt arise in part because of differences in model projections of

future climate or in model speciation of $PM_{2.5}$. In this section, we investigate whether differences in model representation of the sensitivity of $PM_{2.5}$ to meteorological variability may also contribute to uncertainty in projections of future $PM_{2.5}$.

Page 12, line 6-8. This section consists of two parts. First, we test the capability of four ACCMIP models and GEOS-Chem in capturing the observed relationship between JJA monthly mean $PM_{2.5}$ and temperature. We find no model simulates this relationship well. Second, using GEOS-Chem as a testbed, we investigate the reasons of this failure in this particular model.

*- One topic I would encourage the authors to discuss further in their manuscript is the impacts of 2050 climate derived from the CMIP5 ensemble and the regression model. Only ensemble-mean results are presented. (a) I would be very interested in seeing the differences in the projections from individual GCMs. (b) Additionally, I would like to see the interannual variability across individual years in the 20-yr penalty estimates. How robust is the study's finding of an anthropogenic-induced climate change impact by 2050 under a stabilization scenario, given that natural variability has been shown to significantly influence U.S. temperature and precipitation projections on timescales as long as 50 years (Deser et al., 2014; doi: doi:10.1175/JCLI-D-13-00451.1.)? There is a great opportunity to explore climate-related uncertainty in PM2.5 projections at much greater depth within this data.*

**Response**: These are two excellent questions.
In response, we have added several new figures in the supplement. The first shows the distributions of predicted changes of $PM_{2.5}$ concentrations for each CMIP5 models. We also refer to this Figure in the main text.

Page 10, line 8-10. In general, these models agree on the sign of the change of $PM_{2.5}$ across the East by 2050s, but the magnitude of the change varies among models (Figure S5).

Effects of 2050s climate change on annual mean PM$_{2.5}$ in each CMIP5 model

[Figure]

**Figure S5**. Effects of climate change from 2000-2019 to 2050-2069 on annual mean PM$_{2.5}$ concentrations, calculated with observed relationships of PM$_{2.5}$ and meteorology and with meteorology projected by each of the 19 CMIP5 models. White areas denote the regions with no PM$_{2.5}$ observations. For those models providing an ensemble of simulations for the RCP4.5 scenario, only one simulation was chosen for application to our model.

We have also plotted the timeseries of PM$_{2.5}$ changes as annual, summertime, and wintertime means across eight regions from 2000 to 2069 (Figure S9-11). These figures have been added to the Supplement and are referred in the main text.

Page 10, line 14-19. We also examine the 2000-2069 timeseries of projected PM$_{2.5}$ concentrations as annual, summertime, and wintertime means, averaged over eight different U.S. (Figure S8-11). The spread in PM$_{2.5}$ trends is one measure of the uncertainty in our projections, arising in part from differences in model sensitivity to changing greenhouse gases and in part from internal variability of the climate system (e.g., Deser et al., 2013). Averaging results across

the CMIP5 ensemble reveals a robust response of $PM_{2.5}$ to increasing greenhouse gases, at least in some regions, giving us confidence in our approach.

[Figure]

**Figure S8**. The eight U.S. geographical regions used for Figures S9-11.

[Figure]

**Figure S9**. Regional trends in annual mean $PM_{2.5}$ concentrations from 2000 to 2069, calculated with observed relationships of $PM_{2.5}$ and meteorology and with meteorology projected by an

ensemble of 19 CMIP5 models. Shading denotes one standard deviation the mean change across models. The slopes of the timeseries over the 70-yr timeframe are shown inset. Figure S8 defines the eight regions.

[Figure]

**Figure S10**. Similar as Figure S9, but for trend in JJA PM$_{2.5}$ concentrations.

[Figure]

**Figure S11.** Similar as Figure S9, but for trends in DJF PM$_{2.5}$ concentrations.

To elucidate the influence from the internal variability of climate system, we need to analyze numerous simulations with one single model, as inferred from Deser et al. (2013). But here we only use the simulated meteorology from the first ensemble run for each model, so the sources of the uncertainty shown in Figure S9-11 consist of two parts, including the internal variability of climate system as well as the spread of physical parameterization among different models. Since the investigation into the influence of natural variability takes a lot of efforts and it is outside the scope of this study, we decide to briefly discuss this subject and leave the more detailed analysis to the follow-up study. Thanks for making such a good suggestion.

We now bring up the issue of internal variability of the climate system in the Discussion section.

Page 15, line 9-12. Drawbacks of this study include its assumption of constant anthropogenic emissions and its dependence on a relative short history (~15 years) of PM$_{2.5}$ observations. We also do not explicitly consider the role of interannual variability in the climate system and how that might influence our results (Deser et al., 2013).

*- I suggest combining figures 4 and 5 and showing additional details of the penalty projections within the manuscript.*

**Response**: We have combined Figure 4 and 5.

[Figure]

**Figure 4**. Effects of climate change from 2000-2019 to 2050-2069 on (a-d) seasonal and (e) annual mean PM$_{2.5}$ concentrations, calculated with observed relationships of PM$_{2.5}$ and meteorology and with meteorology projected by an ensemble of 19 CMIP5 models. The panels show the mean change in surface PM$_{2.5}$, averaged across the projections. White areas refer to the regions with no PM$_{2.5}$ observations or with fewer than 14 models yielding the same sign of changes.

*- One analysis that is absent but would greatly benefit the study is a comparison of a climate penalty projection generated by the regression model to that generated with a CTM for the same GCM meteorological fields. Comparing the midcentury climate penalty estimated with the regression model, to a projection generated by driving a CTM with the same weather fields (e.g. a GEOS-Chem simulation driven by the present/future met fields from one of the CMIP5 models) would provide great insight into the potential to replace computationally expensive CTMs with a statistical model, and limitations associated with either approach. I encourage the authors to undertake this analysis in future work.*

**Response**:  This is an excellent suggestion for future work. We have added this idea to the Discussion section.
Page 15, line 13-15.  Within these limitations, this study ….. It also demonstrates the utility of a computationally efficient model whose projections of the climate penalty on air quality can be readily compared to those from more traditional dynamic models.

*- The penalty projections generated using the regression model assume that observed relationships between PM2.5 and meteorology remain valid at midcentury under significantly*

*different meteorological conditions and emissions levels. Is this an adequate assumption? An interesting analysis would be to compare the penalty projections of regression models generated under different levels of emissions within the 15-yr observational record.*

**Response**: We have added one figure in the supplement to show the slopes of JJA $PM_{2.5}$ as well as its components with temperature for 1999-2006 and 2007-2013. We also discuss the influence of changing emissions on the sensitivity of $PM_{2.5}$ to climate change.

Page 11, line 14-24. One weakness of this study is that when estimating the sensitivity of $PM_{2.5}$ to meteorological variables, we do not consider the impact of changing anthropogenic emissions on this sensitivity. Figure S13 compares the slopes of monthly mean $PM_{2.5}$ and its components with temperature for two time periods: 1999-2006 summers with high anthropogenic emissions and 1997-2013 summers with low anthropogenic emissions. Using the monthly data, we find that the changes of sensitivity of $PM_{2.5}$ to temperature vary across different locations and species. As the anthropogenic emissions decrease, the slopes of $PM_{2.5}$ and temperature decrease over the Great Plains and Midwest, but increase slightly in the south Atlantic States. Sulfate exhibits decreased sensitivity across the eastern United States, and OC shows no significant pattern of change. Reasons for such inconsistencies may be related to the shorter time periods and therefore less robust sensitivity. In this study, we have thus chosen not to investigate the influence of changing emissions on the sensitivity of $PM_{2.5}$ to climate change using this statistical model.

[Figure]

**Figure S13**. The slopes of detrended (a-b) monthly mean $PM_{2.5}$ and (c-j) different $PM_{2.5}$ components with surface air temperature for 1999-2013 summer months. Left column shows slopes for 1999-2006 with relatively high NOx emissions, and right column shows slopes for 2007-2013 with relatively low NOx emissions. Organic aerosol (OA) in Panel (e-f) is inferred from the measured organic carbon (OC) component using an OA/OC mass ratio of 1.8 (Canagaratna et al., 2015). White areas indicate either missing data or grid boxes where the slope is not significant at the 0.10 level. We note that the observation network has fewer sites in 1999 and 2000 than more recent years.

*- The study explores PM2.5 response to surface temp in 4 CCMs, and then further investigates the dependency in GEOS-Chem. Given the differences between GEOSChem (a CTM) and CCMs, what insights from the GEOS-Chem analysis may useful to identify the causes for the discrepancies between observed and simulated sensitivities in CCMs? Would the authors expect to see any similarities?*

**Response**:  The reviewer raises an interesting question: is it possible that a CTM with specified meteorology calculates a different $PM_{2.5}$ sensitivity to temperature than CCMs with interactive chemistry and climate?  Answering that question is beyond the scope of this paper as it would require a suite of sensitivity studies with CCMs. Below we iterate our response to a similar question from Reviewer 1.

Page 14, line 1-3.  With regard to the ACCMIP results, understanding the failure of these models to capture the observed slopes of monthly mean total $PM_{2.5}$ and temperature is beyond the scope of this paper. Key diagnostics, such as the production rates of sulfate through different oxidation pathways, are not available.

*- PM2.5 concentrations and meteorology could be mapped on a finer resolution grid, and CMIP5 fields interpolated onto that grid. Would there be a benefit or significant change if the statistical regression model were built at higher resolution, rather than the coarse 2.5◦x2.5◦?*

**Response**: This is an interesting suggestion. For this study, we use $2.5° \times 2.5°$ horizontal resolution because this is the resolution in the NCEP Reanalysis 1, and most of the CMIP5 models have comparable resolution. We agree that mapping onto a finer resolution grid could be useful, especially for projections of air quality in urban areas. However, simple interpolation of CMIP5 meteorological fields could lead to large uncertainty and statistically downscaled fields are currently available for only a few CMIP5 variables (temperature and precipitation).

For now we choose not to speculate on the outcome of using finer spatial resolution in our regression model.

*- When listing the range of reported projections for climate change impacts on PM2.5 (e.g. pg. 3, line 7), I recommend using the updated range from the reviews published by Fiore et al. (doi: 10.1080/10962247.2015.1040526.)*

**Response**: We checked the range reported in Fiore et al. (2015) and have updated the text.

Page 3, Line 7. Reviewing earlier studies, Jacob and Winner (2009) and Fiore et al. (2015) concluded that the most of the projected effects of 21[st] century climate changes on $PM_{2.5}$ concentrations are in the range of $\pm 0.1$-1 µg m[-3], with changes up to $\pm 2$ µg m[-3] in certain seasons or regions.